# Accelerating Quantum Reinforcement Learning
# with a Quantum Natural Policy Gradient Based Approach

**Yang Xu** [1]  **Vaneet Aggarwal** [1]

## Abstract

We address the problem of quantum reinforcement learning (QRL) under model-free settings with quantum oracle access to the Markov Decision Process (MDP). This paper introduces a Quantum Natural Policy Gradient (QNPG) algorithm, which replaces the random sampling used in classical Natural Policy Gradient (NPG) estimators with a deterministic gradient estimation approach, enabling seamless integration into quantum systems. While this modification introduces a bounded bias in the estimator, the bias decays exponentially with increasing truncation levels. This paper demonstrates that the proposed QNPG algorithm achieves a sample complexity of $\tilde{\mathcal{O}}(\epsilon^{-1.5})$ for queries to the quantum oracle, significantly improving the classical lower bound of $\tilde{\mathcal{O}}(\epsilon^{-2})$ for queries to the MDP.

## 1. Introduction

Reinforcement Learning (RL) is a foundational framework for sequential decision-making, with applications spanning robotics, finance, transportation, and healthcare (Al-Abbasi et al., 2019; Gonzalez et al., 2023; Tamboli et al., 2024; Wang et al., 2023). The goal of an RL agent is to derive a policy that maximizes the discounted sum of expected rewards through interactions with the environment. A prominent approach to solving this problem is the policy gradient (PG) method, which optimizes directly in the policy space. In classical RL, the optimal sample complexity of this approach has been established as $\tilde{\mathcal{O}}(\epsilon^{-2})$ (Mondal & Aggarwal, 2024a). Quantum computing has shown potential for speedups in areas such as mean estimation (Hamoudi, 2021), multi-armed bandits (Wan et al., 2023; Wu et al., 2023), and tabular reinforcement learning (Ganguly et al.,

2025; Zhong et al., 2024). This work aims to explore potential quantum speedups for reinforcement learning with general parameterized policies.

A key enabler of quantum speedups is the quantum evaluation oracle, which leverages quantum parallelism, amplitude amplification, and entanglement. This powerful concept has been utilized in foundational algorithms such as Grover's search (Grover, 1996). In this work, we assume access to a quantum transition oracle and a quantum initial state oracle, which serve as quantum analogs for sampling from the classical environment and the initial state distribution, respectively. Using these quantum oracles, we aim to demonstrate the speedups that quantum computing can provide in the context of reinforcement learning.

Quantum computing principles have shown significant improvements in areas such as quantum mean estimation (Hamoudi, 2021) and quantum stochastic convex optimization (Sidford & Zhang, 2024). However, quantum reinforcement learning (RL) presents unique challenges. Unlike classical algorithms, quantum methods rely on deterministic operations, complicating the design of unbiased estimators for policy gradients. This paper addresses the trade-off between bias and unbiased estimation by introducing a novel deterministic algorithm that facilitates quantum-compatible gradient estimation. While this approach introduces a bounded bias, it achieves orders reduction in sample complexity compared to its classical counterparts.

### 1.1. Challenges and Contributions

In this work, we introduce the first quantum model-free reinforcement learning (RL) algorithm with theoretical guarantees, addressing the challenges posed by large state and action spaces. To our knowledge, this is the first work to coherently embed the entire Natural Policy Gradient (NPG) into a quantum state by leveraging only the standard environment oracles from reinforcement learning. This novel construction allows subsequent quantum subroutines to utilize these NPG gradients directly in superposition, enabling potential accelerations in policy optimization. Extending classical model-free RL algorithms to the quantum domain is nontrivial. Specifically, classical model-free algorithms (Mondal & Aggarwal, 2024a; Liu et al., 2020; Agarwal

[1]Purdue University, West Lafayette, IN, USA. Correspondence to: Yang Xu <xu1720@purdue.edu>, Vaneet Aggarwal <vaneet@purdue.edu>.

*Proceedings of the $42^{nd}$ International Conference on Machine Learning*, Vancouver, Canada. PMLR 267, 2025. Copyright 2025 by the author(s).

et al., 2021) typically rely on sampling trajectories of random lengths following a geometric distribution for policy gradient estimation. However, encoding such trajectories with random lengths into quantum systems is challenging. To address this, we propose a novel deterministic sampling algorithm that uses truncation to estimate both the Fisher matrix and the policy gradient. We further develop a mini-batch strategy to incorporate quantum variance reduction into the Natural Policy Gradient (NPG) update. By carefully analyzing the bias introduced by the truncation, we show that our approach achieves a sample complexity of $\tilde{\mathcal{O}}(\epsilon^{-1.5})$, surpassing the classical lower bound of $\tilde{\mathcal{O}}(\epsilon^{-2})$. To the best of our knowledge, this is the first work to demonstrate quantum speedups for parameterized model-free infinite-horizon Markov Decision Processes (MDPs).

It should be noted that SGD-based approaches have been shown to achieve a sample complexity of $\tilde{\mathcal{O}}(\epsilon^{-1.5})$ for stochastic convex optimization (Sidford & Zhang, 2024). However, reinforcement learning (RL) presents a fundamentally different set of challenges compared to stochastic convex optimization. While insights from the literature suggest that $\tilde{\mathcal{O}}(\epsilon^{-1.5})$ is likely the best achievable sample complexity for SGD-based approaches, this limitation arises because quantum computing can accelerate the inner loop but not the outer loop of such algorithms. To achieve better sample complexity guarantees, stochastic cutting-plane methods have been proposed in (Sidford & Zhang, 2024). Extending these methods to RL, however, would require an entirely new algorithmic framework. This is due to the non-convex nature of RL with general parameterization, which makes direct adaptation infeasible. Moreover, the techniques used in (Agarwal et al., 2021) to transition from local to global convergence are not applicable in this context, preventing similar guarantees for such methods in RL.

## 1.2. Contributions

The main contributions of this paper are summarized as follows:

- We construct a quantum oracle for NPG gradient estimation by leveraging fundamental oracles from the Markov Decision Process (MDP), ensuring efficient computation in quantum environments.

- We modify the classical Natural Policy Gradient (NPG) algorithm into a deterministic setting, enabling seamless integration into quantum systems while maintaining bounded gradient estimation bias.

- Our proposed approach achieves a sample complexity of $\tilde{O}(\epsilon^{-1.5})$, which surpasses the classical lower bound of $\tilde{O}(\epsilon^{-2})$, demonstrating a significant improvement in efficiency.

## 1.3. Related Work and Preliminaries

**Quantum Mean Estimation:** Mean estimation focuses on determining the average value of samples drawn from an unknown probability distribution. Notably, quantum mean estimation provides a quadratic improvement over classical methods (Montanaro, 2015; Hamoudi, 2021). This enhancement arises from the utilization of quantum amplitude amplification, which facilitates the suppression of undesired quantum states relative to the target states to be extracted (Brassard et al., 2002).

In what follows, we define the key concepts and results related to quantum mean estimation that are central to our analysis. Specifically, we introduce the definition of a classical random variable alongside the quantum sampling oracle, which is used to perform quantum experiments.

**Definition 1** (Random Variable, Definition 2.2 of (Cornelissen et al., 2022))**.** A finite random variable can be represented as $X : \Omega \to E$ for some probability space $(\Omega, \mathbb{P})$, where $\Omega$ is a finite sample set, $\mathbb{P} : \Omega \to [0, 1]$ is a probability mass function and $E \subset \mathbb{R}$ is the support of $X$. $(\Omega, \mathbb{P})$ is frequently omitted when referring to the random variable $X$.

To perform quantum mean estimation, we provide the definition of quantum experiment. This is analogous to classical random experiments.

**Definition 2** (Quantum Experiment)**.** Consider a random variable $X$ on a probability space $(\Omega, 2^\Omega, \mathbb{P})$. Let $\mathcal{H}_\Omega$ be a Hilbert space with basis states $\{|\omega\rangle\}_{\omega \in \Omega}$ and fix a unitary $\mathcal{U}_\mathbb{P}$ acting on $\mathcal{H}_\Omega$ such that

$$\mathcal{U}_\mathbb{P} : |0\rangle \mapsto \sum_{\omega \in \Omega} \sqrt{\mathbb{P}(\omega)}|\omega\rangle$$

assuming $0 \in \Omega$. We define a *quantum experiment* as the process of applying the unitary $\mathcal{U}_\mathbb{P}$ or its inverse $\mathcal{U}_\mathbb{P}^{-1}$ on any state in $\mathcal{H}_\Omega$.

The unitary operator $\mathcal{U}_\mathbb{P}$ enables the query of samples of the random variable in superposition, forming the basis for the speed-ups achieved in quantum algorithms. To perform quantum mean estimation for random variable values (Cornelissen et al., 2022; Hamoudi, 2021; Montanaro, 2015), the implementation of an additional quantum evaluation oracle is required.

**Definition 3** (Quantum Evaluation Oracle)**.** Consider a finite random variable $X : \Omega \to E$ on a probability space $(\Omega, 2^\Omega, \mathbb{P})$. Let $\mathcal{H}_\Omega$ and $\mathcal{H}_E$ be two Hilbert spaces with basis states $\{|\omega\rangle\}_{\omega \in \Omega}$ and $\{|x\rangle\}_{x \in E}$ respectively. We say that a unitary $\mathcal{U}_X$ acting on $\mathcal{H}_\Omega \otimes \mathcal{H}_E$ is a quantum evaluation oracle for $X$ if

$$\mathcal{U}_X : |\omega\rangle|0\rangle \mapsto |\omega\rangle|X(\omega)\rangle$$

for all $\omega \in \Omega$, assuming $0 \in E$.

Next, we present a key quantum mean estimation result that will be carefully employed in our algorithmic framework to utilize the quantum superpositioned states collected by the RL agent. One of the crucial aspects of Lemma 1 is that quantum mean estimation converges at the rate $\mathcal{O}(\frac{1}{n})$ as opposed to classical benchmark convergence rate $\mathcal{O}(\frac{1}{\sqrt{n}})$ for $n$ number of samples, therefore estimation efficiency increases quadratically.

**Lemma 1** (Quantum multivariate bounded estimator, Theorem 3.5 of (Cornelissen et al., 2022))**.** *Given access to the quantum sampling oracle $\mathcal{U}_X$ and let $d$ be the dimension of $X$, for any $\hat{\sigma}, \delta \geq 0$ there is a procedure* QuantumMeanEstimation$(X, \hat{\sigma}, \delta)$ *that uses $\tilde{\mathcal{O}}(L\sqrt{d}\log(1/\delta)/\hat{\sigma})$ queries and outputs an estimate $\hat{\mu}$ of the expectation $\mu$ of any $d$-dimensional random variable $X$ satisfying $Var[X] \leq L^2$ with error $\|\hat{\mu} - \mu\| \leq \hat{\sigma} \leq L$ and success probability $1 - \delta$. Furthermore, if $d < L\sqrt{d}\log(1/\delta)/\hat{\sigma}$,* QuantumMeanEstimation$(X, \hat{\sigma}, \delta)$ *can be implemented using Algorithm 2 in (Cornelissen et al., 2022).*

**Quantum Reinforcement Learning:** In recent years, Quantum Reinforcement Learning (QRL) has garnered substantial interest from the research community (Dong et al., 2008; Paparo et al., 2014; Dunjko et al., 2017; Jerbi et al., 2021). Within this field, studies on the Quantum Multi-Armed Bandits (Q-MAB) problem have demonstrated exponential reductions in regret by leveraging amplitude amplification techniques (Wang et al., 2021b; Casalé et al., 2020). However, the theoretical foundations of Q-MAB approaches are not directly applicable to QRL, primarily due to the lack of state transitions in bandit problems.

Although interest in QRL has been growing, most existing works are still limited to the tabular setup. For instance, (Zhong et al., 2024; Ganguly et al., 2025) showed that logarithmic regret is achievable in episodic/average reward Markov Decision Processes (MDPs) using model-based approaches, while (Wang et al., 2021a) investigated the infinite horizon discounted MDP framework with tabular model-free setups. However, the results don't directly extend to large state spaces, where a parameterized policy is used. Further, the problem becomes non-convex due to the use of general parameterization. To the best of our knowledge, ours is the first work to address infinite horizon MDPs with general parameterized policies.

## 2. Formulation

### 2.1. Quantum Computing Basics:

In this section, we provide a concise overview of the key concepts relevant to our work, drawing from (Nielsen & Chuang, 2010) and (Wu et al., 2023). Let $\mathbb{C}^m$ denote an $m$-dimensional Hilbert space. A quantum state $|x\rangle = $ $(x_1, \ldots, x_m)^T$ can be represented as a vector within this space, satisfying the normalization condition $\sum_i |x_i|^2 = 1$. For a finite set $\xi = \{\xi_1, \ldots, \xi_m\}$ of $m$ elements, each $\xi_i \in \xi$ can be mapped to an element of an orthonormal basis in $\mathbb{C}^m$ as follows:

$$\xi_i \mapsto |\xi_i\rangle \equiv e_i, \tag{1}$$

where $e_i$ represents the $i$th unit vector in $\mathbb{C}^m$. Utilizing this notation, any arbitrary quantum state $|x\rangle = (x_1, \ldots, x_m)^T$ can be expressed in terms of the elements of $\xi$ as:

$$|x\rangle = \sum_{n=1}^m x_n|\xi_n\rangle, \tag{2}$$

where $|x\rangle$ is a quantum superposition of the basis states $|\xi_1\rangle, \ldots, |\xi_m\rangle$, and $x_n$ denotes the amplitude corresponding to $|\xi_n\rangle$. To extract classical information from the quantum state, a measurement is performed, causing the state to collapse to one of the basis states $|\xi_i\rangle$ with probability $|x_i|^2$. In quantum computing, quantum states are typically represented using input or output registers composed of qubits, which may exist in superposition.

### 2.2. MDP Formulation

We analyze a Markov Decision Process (MDP) represented by the tuple $\mathcal{M} = (\mathcal{S}, \mathcal{A}, r, P, \gamma, \rho)$, where $\mathcal{S}$ and $\mathcal{A}$ denote the state space and action space, respectively. The reward function $r : \mathcal{S} \times \mathcal{A} \to [0, 1]$ assigns a reward to each state-action pair, and the state transition kernel $P : \mathcal{S} \times \mathcal{A} \to \Delta^{|\mathcal{S}|}$ defines the probabilities of transitioning between states (with $\Delta^{|\mathcal{S}|}$ denoting the probability simplex over $|\mathcal{S}|$ states). The discount factor $\gamma \in (0, 1)$ determines the importance of future rewards, while $\rho \in \Delta^{|\mathcal{S}|}$ specifies the initial state distribution. A policy $\pi : \mathcal{S} \to \Delta^{|\mathcal{A}|}$ provides the probability distribution over actions for a given state. The $Q$-function for a policy $\pi$, corresponding to a state-action pair $(s, a)$, is defined as follows:

$$Q^\pi(s, a) \triangleq \mathbb{E}\left[\sum_{t=0}^\infty \gamma^t r(s_t, a_t) \bigg| s_0 = s, a_0 = a\right]$$

Here, the expectation is taken over all trajectories $\{(s_t, a_t)\}_{t=0}^\infty$ induced by $\pi$, where $s_t \sim P(s_{t-1}, a_{t-1})$ and $a_t \sim \pi(s_t)$ for all $t \geq 1$. Similarly, the $V$-function associated with the policy $\pi$ is defined as:

$$V^\pi(s) \triangleq \mathbb{E}\left[\sum_{t=0}^\infty \gamma^t r(s_t, a_t) \bigg| s_0 = s\right] = \sum_{a \in \mathcal{A}} \pi(a|s)Q^\pi(s, a)$$

The advantage function is then given by:

$$A^\pi(s, a) \triangleq Q^\pi(s, a) - V^\pi(s), \ \forall(s, a) \in \mathcal{S} \times \mathcal{A}$$

The primary objective is to maximize the following function over all possible policies:

$$J_\rho^\pi \triangleq \mathbb{E}_{s \sim \rho}[V^\pi(s)] = \frac{1}{1 - \gamma} \sum_{s,a} d_\rho^\pi(s, a)\pi(a|s)r(s, a)$$

Here, the state occupancy $d_\rho^\pi \in \Delta^{|\mathcal{S}|}$ is defined as:

$$d_\rho^\pi(s) \triangleq (1-\gamma)\sum_{t=0}^\infty \gamma^t \mathrm{Pr}(s_t = s|s_0 \sim \rho, \pi), \ \forall s \in \mathcal{S}$$

The state-action occupancy $\nu_\rho^\pi \in \Delta^{|\mathcal{S}\times\mathcal{A}|}$ is similarly defined as $\nu_\rho^\pi(s,a) \triangleq d_\rho^\pi(s)\pi(a|s)$ for all $(s,a) \in \mathcal{S} \times \mathcal{A}$. In practical applications, the size of the state space often necessitates parameterizing policies using deep neural networks with d-dimensional parameters. Let $\pi_\theta$ represent a policy parameterized by $\theta \in \mathbb{R}^d$. Under this parameterization, the optimization problem can be reformulated as:

$$\max_{\theta \in \mathbb{R}^d} \ J_\rho^{\pi_\theta} \tag{3}$$

For simplicity, we denote $J_\rho^{\pi_\theta}$ as $J_\rho(\theta)$ throughout the remainder of this paper.

### 2.3. Quantum access to an MDP

We adopt the framework and notations of quantum computing from (Zhong et al., 2024; Wiedemann et al., 2022; Ganguly et al., 2025; Jerbi et al., 2023) to design the quantum sampling oracle for reinforcement learning (RL) environments, enabling the modeling of an agent's interactions with an unknown MDP environment. We proceed to define quantum-accessible RL environments corresponding to the classical MDP $\mathcal{M}$. For an agent at step $t$ in state $s_t$ and performing action $a_t$, we construct quantum sampling oracles for the transition probabilities of the next state, $P(\cdot|s_t, a_t)$. To this end, let two Hilbert spaces, $\bar{\mathcal{S}} = \mathbb{C}^{|\mathcal{S}|}$ and $\bar{\mathcal{A}} = \mathbb{C}^{|\mathcal{A}|}$, represent the superpositions of classical states and actions, respectively. The computational bases for these Hilbert spaces are denoted as $\{|s\rangle\}_{s\in\mathcal{S}}$ and $\{|a\rangle\}_{a\in\mathcal{A}}$. We assume the ability to implement the following quantum sampling oracles:

- Quantum transition oracle $\mathcal{U}_P$: The quantum evaluation oracle for the transition probability (quantum transition oracle) $\mathcal{U}_P$ which at step $t$, returns the superposition over $s' \in \mathcal{S}$ according to $P(s'|s_t, a_t)$, the probability distribution of the next state given the current state $|s_t\rangle$ and action $|a_t\rangle$ is defined as:

$$\mathcal{U}_P : |s_t\rangle|a_t\rangle|0\rangle \to |s_t\rangle|a_t\rangle \sum_{s'\in\mathcal{S}} \sqrt{P(s'|s_t,a_t)}|s'\rangle \tag{4}$$

- Quantum initial state oracle $\mathcal{U}_\rho$: The quantum sampling of the initial state distribution (quantum initial state oracle) $\mathcal{U}_\rho$ which when queried, returns a superposition over $s \in \mathcal{S}$ according to $\rho$ is defined as:

$$\mathcal{U}_\rho : |0\rangle \to \sum_{s\in\mathcal{S}} \sqrt{\rho(s)}|s\rangle \tag{5}$$

We also assume the ability to construct a unitary $\Pi$ that coherently implements a policy $\pi_\theta$:

- Let $\pi_\theta : \mathcal{S}\times\mathcal{A} \to [0,1]$ be a reinforcement learning policy acting in a state-action space $\mathcal{S} \times \mathcal{A}$ and parametrized by a vector $\theta \in \mathbb{R}^d$ (that can be encoded with finite precision as $|\theta\rangle$). We say that the policy is quantum-evaluatable if we can construct a unitary satisfying:

$$\Pi : |\theta\rangle|s\rangle|0\rangle \mapsto |\theta\rangle|s\rangle \sum_{a\in\mathcal{A}} \sqrt{\pi_\theta(a|s)}|a\rangle. \tag{6}$$

This construction is particularly intuitive for certain quantum policies, such as RAW-PQC which is introduced in (Jerbi et al., 2023). However, any policy that can be classically computed can also be converted into such a unitary operation through quantum simulation of the classical computation of $(\pi_\theta(a|s) : a \in \mathcal{A})$ and leveraging established methods to encode this probability vector into the amplitudes of a quantum state (Grover & Rudolph, 2002). With access to quantum oracles for both the environment and the policy in (4)-(6), it becomes possible to design subroutines capable of generating superpositions of trajectories with fixed length within the environment and calculating the returns associated with these trajectories. We can further utilize $\mathcal{U}_P, \mathcal{U}_\rho$ and $\Pi$ defined above to construct a unitary $\mathcal{U}_{P(\tau_N)}$ as follows: Let $\mathcal{M}$ be a quantum-accessible MDP with oracles $\mathcal{U}_P, \mathcal{U}_\rho$ as above, and let $\pi_\theta$ be a quantum-evaluatable policy with its unitary implementation $\Pi$ as defined above. Given a fixed number $N$, the unitary $\mathcal{U}_{P(\tau_N)}$ that prepares a coherent superposition of all trajectories $\tau_N = (s_0, a_0, \ldots, s_{N-1}, a_{N-1})$ of length $N$ (without their rewards) is defined as ,

$$\mathcal{U}_{P(\tau_N)} : |\theta\rangle|0\rangle^{\otimes 2N} \mapsto |\theta\rangle \sum_{\tau_N} \sqrt{P_\theta(\tau_N)}|\tau_N\rangle \tag{7}$$

for $P_\theta(\tau_N) = \rho(s_0)\prod_{t=0}^{N-1} \pi_\theta(a_t|s_t)P(s_{t+1}|s_t, a_t)$. The details of constructing $\mathcal{U}_{P(\tau_N)}$ using $\mathcal{U}_P, \mathcal{U}_\rho$ and $\Pi$ are characterized in Appendix A.

## 3. Proposed Algorithm

A common way to solve the maximization (3) in classical methods is via updating the policy parameters by applying the gradient ascent: $\theta_{k+1} = \theta_k + \eta\nabla_\theta J_\rho(\theta_k)$, $k \in \{0, 1, \cdots\}$ starting with an initial parameter, $\theta_0$. Here, $\eta > 0$ denotes the learning rate, and the policy gradients (PG) are given as follows (Sutton et al., 1999).

$$\nabla_\theta J_\rho(\theta) = \frac{1}{1-\gamma}\mathbb{E}_{(s,a)\sim\nu_\rho^{\pi_\theta}}\left[A^{\pi_\theta}(s,a)\nabla_\theta \log \pi_\theta(a|s)\right] \tag{8}$$

This paper uses the natural policy gradient (NPG) to update the policy parameters. In particular, $\forall k \in \{1, 2, \cdots\}$, we

**Algorithm 1** Quantum Natural Policy Gradient

---

1: **Input:** Policy Parameter $\theta_0$, State distribution $\rho$, Runtime Parameters $K, H, N, \hat{\sigma}_g, \hat{\sigma}_F$, Learning Parameters $(\eta, \alpha)$, NPG initialization $\omega_0$

2: **for** $k \leftarrow \{0, \cdots, K-1\}$ **do**

3:    $\mathbf{x}_0, \mathbf{v}_0 \leftarrow \mathbf{0}$

4:    **for** $h \in \{0, \cdots, H-1\}$ **do**

5:       {Mini-Batch Quantum SGD}

6:       $\tilde{g}_h \leftarrow \texttt{QVarianceReduce}(\hat{g}_\rho(\tau_N|\theta_k), \hat{\sigma}_g^2)$ Obtain gradient estimator

7:       $\tilde{F}_h \leftarrow \texttt{QVarianceReduce}(\hat{F}_\rho(\tau_N|\theta_k), \hat{\sigma}_F^2)$ Obtain Fisher estimator

8:       $\nabla_\omega \tilde{L}(\omega, \tau_N) \leftarrow \tilde{F}_h \omega - \tilde{g}_h$

9:       NPG update: $\omega_{h+1} = \omega_h - \alpha \nabla_\omega \tilde{L}(\omega, \tau_N)\big|_{\omega=\omega_h}$

10:    **end for**

11:    $\omega_k = \omega_H$

12:    Policy Parameter Update:

$$\theta_{k+1} \leftarrow \theta_k + \eta \omega_k \qquad (11)$$

13: **end for**

14: **Output:** $\{\theta_k\}_{k=0}^{K-1}$

---

have,

$$\theta_{k+1} = \theta_k + \eta F_\rho(\theta_k)^\dagger \nabla_\theta J_\rho(\theta_k) \qquad (9)$$

where $\dagger$ is the Moore-Penrose pseudoinverse operator, and $F_\rho$ is called the Fisher information matrix which is defined as follows $\forall \theta \in \mathbb{R}^d$.

$$F_\rho(\theta) \triangleq \mathbb{E}_{(s,a) \sim \nu_\rho^{\pi_\theta}} \left[ \nabla_\theta \log \pi_\theta(a|s) \otimes \nabla_\theta \log \pi_\theta(a|s) \right] \qquad (10)$$

where $\otimes$ denotes the outer product. In particular, for any $\mathbf{a} \in \mathbb{R}^d, \mathbf{a} \otimes \mathbf{a} = \mathbf{a}\mathbf{a}^{\mathrm{T}}$.

Let $\omega_\theta^* \triangleq F_\rho(\theta)^\dagger \nabla_\theta J_\rho(\theta)$. Notice that we have removed the dependence of $\omega_\theta^*$ on $\rho$ for notational convenience. Invoking (8), the term $\omega_\theta^*$ can be written as a solution of a quadratic optimization (Peters & Schaal, 2008). Specifically, $\omega_\theta^* = \arg\min_\omega L_{\nu_\rho^{\pi_\theta}}(\omega, \theta)$ where $L_{\nu_\rho^{\pi_\theta}}(\omega, \theta)$ is the compatible function approximation error and it is mathematically defined as,

$$L_{\nu_\rho^{\pi_\theta}}(\omega, \theta) \triangleq \frac{1}{2} \mathbb{E}_{(s,a) \sim \nu_\rho^{\pi_\theta}} \left[ \frac{1}{1-\gamma} A^{\pi_\theta}(s,a) \right. $$
$$\left. -\omega^{\mathrm{T}} \nabla_\theta \log \pi_\theta(a|s) \right]^2 \qquad (12)$$

Using the above notations, NPG updates can be written as $\theta_{k+1} = \theta_k + \eta \omega_k^*$, $k \in \{1, 2, \cdots\}$ where $\omega_k^* = \omega_{\theta_k}^*$. However, in most practical scenarios, the learner does not

have knowledge of the state transition probabilities, making it difficult to directly calculate $\omega_k^*$. In the following, we clarify how one can get its sample-based estimates. Note that the gradient of $L_{\nu_\rho^{\pi_\theta}}(\cdot, \theta)$ can be obtained as shown below for any arbitrary $\theta$.

$$\nabla_\omega L_{\nu_\rho^{\pi_\theta}}(\omega, \theta) = F_\rho(\theta)\omega - \nabla_\theta J_\rho(\theta) \qquad (13)$$

We now present an estimation procedure for obtaining quantum estimators for $\nabla_\omega L_{\nu_\rho^{\pi_\theta}}(\omega, \theta)$ based on the provided oracles in Section 2.3. Different from the commonly used unbiased estimators in (Agarwal et al., 2021; Mondal & Aggarwal, 2024a) which requires sampling from geometric distribution with mean $\frac{1}{1-\gamma}$, we construct a deterministic algorithm which can be implemented using the quantum oracles in Section 2.3. For a fixed $N$, let $\tau_N = (s_0, a_0, \ldots, s_N, a_N)$ be a trajectory with length $N$ generated by taking policy $\pi_\theta$ with $s_0$ sampled from $\rho$, we construct the policy gradient estimator $\hat{g}_\rho(\tau_N|\theta)$ and Fisher information matrix estimator $\hat{F}_\rho(\tau_N|\theta)$ as follows:

$$\hat{g}_\rho(\tau_N|\theta) = \sum_{n=0}^{N-1} \left( \sum_{t=0}^n \nabla_\theta \log \pi_\theta(a_t|s_t) \right) (\gamma^n r(s_n, a_n)), \qquad (14)$$

$$\hat{F}_\rho(\tau_N|\theta) = (1-\gamma) \sum_{n=0}^{N-1} \gamma^n \Big( \nabla_\theta \log \pi_\theta(a_n|s_n) \qquad (15)$$
$$\otimes \nabla_\theta \log \pi_\theta(a_n|s_n) \Big),$$

Note that $\hat{g}_\rho(\tau_N|\theta)$ and $\hat{F}_\rho(\tau_N|\theta)$ are truncated estimators deterministically obtained from $\tau_N$. Our goal then is to obtain the above estimators in quantum superpositioned forms from the quantum trajectories in (7). Assume that a quantum state $|\Psi_\theta\rangle = |\theta\rangle \sum_{\tau_N} \sqrt{P_\theta(\tau_N)}|\tau_N\rangle$ represents a superposition of all possible values of $\tau_N$ generated from the MDP under the parameterized policy $\pi_\theta$. We can construct quantum operators $\mathcal{U}_F$ and $\mathcal{U}_g$ such that

$$\mathcal{U}_F\left(|\Psi_\theta\rangle \otimes |0\rangle\right) = |\psi_F^\theta\rangle$$
$$:= |\theta\rangle \sum_{\tau_N} \sqrt{P_\theta(\tau_N)}|\tau_N\rangle \otimes \left|\hat{F}_\rho(\tau_N|\theta)\right\rangle, \qquad (16)$$

$$\mathcal{U}_g\left(|\Psi_\theta\rangle \otimes |0\rangle\right) = |\psi_g^\theta\rangle$$
$$:= |\theta\rangle \sum_{\tau_N} \sqrt{P_\theta(\tau_N)}|\tau_N\rangle \otimes |\hat{g}_\rho(\tau_N|\theta)\rangle, \qquad (17)$$

where $|0\rangle$ is an auxiliary state, $\left|\hat{F}_\rho(\tau_N|\theta)\right\rangle$ and $|\hat{g}_\rho(\tau_N|\theta)\rangle$ represent the quantum states encoding the function value $\hat{F}_\rho(\tau_N|\theta)$ and $\hat{g}_\rho(\tau_N|\theta)$ for each basis state in the superposition $|\tau_N\rangle$. The detailed construction for oracles $\mathcal{U}_F$ and $\mathcal{U}_g$ are described in Appendix A.

Our quantum NPG algorithm is described in Algorithm 1, which can be segregated into a classical *outer loop* and

a novel quantum *inner loop*. In its *outer loop*, the policy parameters are updated $K$ number of times following (11), where $\{\omega_k\}_{k=0}^{K-1}$ denote the estimates of $\{\omega_k^*\}_{k=0}^{K-1}$. For the *inner loop*, the estimates $\{\omega_k\}_{k=0}^{K-1}$ are calculated by iterating the quantum SGD algorithm $H$ number of times. Each quantum SGD iteration consists of a quantum mini-batch approach, where the stochastic gradients are obtained via quantum mean estimation using quantum samples. We utilize `QVarianceReduce` in (Sidford & Zhang, 2024) that leverages `QuantumMeanEstimation` in Lemma 1 as a subroutine. For each $\theta_k$ and each $h$, to perform `QVarianceReduce`, multiple quantum samples for $|\Psi_{\theta_k}\rangle$ are drawn to obtain variance-reduced estimators $\tilde{F}_h$ and $\tilde{g}_h$.

$$\tilde{g}_h = \texttt{QVarianceReduce}(\hat{g}_\rho(\tau_N|\theta), \hat{\sigma}_g^2) \tag{18}$$

$$\tilde{F}_h = \texttt{QVarianceReduce}(\hat{F}_\rho(\tau_N|\theta), \hat{\sigma}_F^2) \tag{19}$$

Note that $\hat{\sigma}_F^2$ and $\hat{\sigma}_g^2$ are the pre-defined target variances, which are specified later in Theorem 3. `QVarianceReduce` offers a quadratic speedup in terms of sample complexity compared with classical mean estimation methods, and the details are specified in Appendix B. Note that in this paper, for random variable $X$ with dimension $d$, we denote the variance of $X$ as the trace of its covariance matrix. With the above quantum variance-reduced estimators, the NPG gradient foor a given policy parameter $\theta$ is calculated and updated as follows:

$$\nabla_\omega \tilde{L}(\omega, \tau_N) = \tilde{F}_h \omega - \tilde{g}_h \tag{20}$$

$$\omega_{h+1} = \omega_h - \alpha \nabla_\omega \tilde{L}(\omega, \tau_N)\big|_{\omega=\omega_h} \tag{21}$$

## 4. Global Convergence Analysis

In this section, we discuss the convergence properties of Algorithm 1. We first present the standard analysis procedures of the classical *outer loop*, and then delve into the results of the quantum *inner loop*, which contributes to the polynomial speedup in the final result.

### 4.1. Outer Loop Analysis

We start with the assumptions commonly used in classical parameterized RL.

**Assumption 1.** The score function is $G$-Lipschitz and $B$-smooth. Mathematically, the following relations hold $\forall \theta, \theta_1, \theta_2 \in \mathbb{R}^d, \forall (s,a) \in \mathcal{S} \times \mathcal{A}$.

$(a)\ \|\nabla_\theta \log \pi_\theta(a|s)\| \leq G$

$(b)\ \|\nabla_\theta \log \pi_{\theta_1}(a|s) - \nabla_\theta \log \pi_{\theta_2}(a|s)\| \leq B\|\theta_1 - \theta_2\|$ $\tag{22}$

where $B$ and $G$ are some positive reals.

The Lipschitz continuity and smoothness of the score function are widely assumed in the literature (Liu et al., 2020; Agarwal et al., 2020), and can be validated for basic parameterized policies, such as Gaussian policies. Under Assumption 1, the following lemma holds.

**Lemma 2.** *If Assumption 1 holds, then $J_\rho(\cdot)$ defined in (3) satisfies the following properties $\forall \theta \in \mathbb{R}^d$.*

$$(a)\ \|\nabla_\theta J_\rho(\theta)\| \leq \frac{G}{(1-\gamma)^2}$$

$$(b)\ J_\rho(\cdot)\ is\ L-smooth,\ L \triangleq \frac{B}{(1-\gamma)^2} + \frac{2G^2}{(1-\gamma)^3}$$

*Proof.* Statement $(a)$ can be proven by combining Assumption 1(a) with (8) whereas statement $(b)$ follows directly from Proposition 4.2 of (Xu et al., 2019). □

Lemma 2 implies the Lipschitz continuity and smoothness for function $J_\rho(\cdot)$, which are critical properties for the further analysis. The following Assumption 2 states that the class of parameterized policies is sufficiently expressive such that, for any policy parameter $\theta$, the error in the transferred compatible function approximation, represented by the left-hand side of equation (23), is bounded by a positive constant, $\epsilon_{\text{bias}}$.

**Assumption 2.** The compatible function approximation error defined in (12) satisfies the following $\forall \theta \in \mathbb{R}^d$.

$$L_{\nu_\rho^{\pi^*}}(\omega_\theta^*, \theta) \leq \epsilon_{\text{bias}} \tag{23}$$

where $\pi^*$ is the optimal policy i.e., $\pi^* = \arg\max_\pi J_\rho^\pi$ and $\omega_\theta^*$ is defined as follows.

$$\omega_\theta^* \triangleq \arg\max_{\omega \in \mathbb{R}^d} L_{\nu_\rho^{\pi_\theta}}(\omega, \theta) \tag{24}$$

It has been shown that for softmax parameterization, $\epsilon_{\text{bias}}$ is equal to zero (Agarwal et al., 2021). Moreover, in the case of expressive neural network-based parameterization, $\epsilon_{\text{bias}}$ is observed to be relatively small (Wang et al., 2019).

**Assumption 3.** There exists a constant $\mu_F > 0$ such that $F_\rho(\theta) - \mu_F I_d$ is positive semidefinite where $I_d$ denotes an identity matrix of dimension d and $F_\rho(\theta)$ is defined in (10). Equivalently, $F_\rho(\theta) \succcurlyeq \mu_F I_d$.

Assumption 3 asserts that the Fisher information function should not be too small, a condition commonly adopted in classical policy gradient literature (Liu et al., 2020; Bai et al., 2023; Zhang et al., 2020). This assumption holds for Gaussian policies with linearly parameterized means. The set of policies that satisfy Assumption 3 is referred to as the Fisher Non-degeneracy (FND) set. It is important to note that $F_\rho(\theta) = \nabla_\omega^2 L_{\nu_\rho^{\pi_\theta}}(\omega, \theta)$. Therefore, Assumption 3 essentially implies that the function $L_{\nu_\rho^{\pi_\theta}}(\cdot, \theta)$ exhibits strong convexity with parameter $\mu_F$. With the above assumption, the following lemma holds:

**Lemma 3** (Corollary 1 of (Mondal & Aggarwal, 2024a)). *Let, the parameters $\{\theta_k\}_{k=0}^{K-1}$ be updated via (11), $\pi^*$ be the optimal policy and $J_\rho^*$ be the optimal value of the function $J_\rho(\cdot)$. If assumptions 1-3 hold, then the following inequality is satisfied for $\eta = \frac{\mu_F^2}{4G^2L}$.*

$$J_\rho^* - \frac{1}{K} \sum_{k=0}^{K-1} \mathbb{E}[J_\rho(\theta_k)]$$

$$\leq \sqrt{\epsilon_{\text{bias}}} + \frac{G}{K} \sum_{k=0}^{K-1} \mathbb{E} \|(\mathbb{E}[\omega_k|\theta_k] - \omega_k^*)\|$$

$$+ \frac{B}{4L} \left( \frac{\mu_F^2}{G^2} + G^2 \right) \left( \frac{1}{K} \sum_{k=0}^{K-1} \mathbb{E} \|\omega_k - \omega_k^*\|^2 \right)$$

$$+ \frac{G^2}{\mu_F^2 K} \left( \frac{B}{1-\gamma} + 4L \, \mathbb{E}_{s \sim d_\rho^{\pi^*}} [KL(\pi^*(\cdot|s)\|\pi_{\theta_0}(\cdot|s))] \right) \tag{25}$$

*where $KL(\cdot\|\cdot)$ is the Kullback-Leibler divergence.*

Lemma 3 decomposes the optimality gap in a way that achieves the order-optimal sample complexity of $\mathcal{O}(\epsilon^{-2})$ in the classical setting. Notice that both the first-order approximation error, $\mathbb{E} \|(\mathbb{E}[\omega_k|\theta] - \omega_k^*)\|$ and the second-order error, $\mathbb{E} \|\omega_k - \omega_k^*\|^2$ are both dependent on $H$, the number of iterations in the *inner loop*.

**4.2. Inner Loop Analysis**

The sample complexity of the quantum *inner loop* is characterized by the total quantum state-action pairs needed in the algorithm, which is equivalent to the total queries of $\mathcal{U}_P$ and $\Pi$. Recall that each iteration in the *inner loop* consists of one operation of (18) and (19), which performs quantum mean estimations of quantum samples $|\psi_F^\theta\rangle$ and $|\psi_g^\theta\rangle$ queried from $\mathcal{U}_F$ and $\mathcal{U}_g$. The complexities of these quantum samples are as follows:

**Lemma 4.** *One call of $\mathcal{U}_F$ and $\mathcal{U}_g$ each requires one call of $\mathcal{U}_\rho$ and $\mathcal{O}(N)$ calls to $\mathcal{U}_P$ and $\Pi$.*

The proof of Lemma 4 is in Appendix A. Lemma 4 states that obtaining a single sample of $|\psi_F^\theta\rangle$ in (16) and $|\psi_g^\theta\rangle$ in (17) requires a sample complexity of $\mathcal{O}(N)$ with respect to the quantum state-action pairs. We next provide the biases and variances for $\hat{g}_\rho(\tau_N|\theta)$ and $\hat{F}_\rho(\tau_N|\theta)$, which are the random variables for the quantum samples $|\psi_F^\theta\rangle$ and $|\psi_g^\theta\rangle$.

**Theorem 1.** *Under assumption 1, for a fixed $\theta$, we have the following results on $\hat{F}_\rho(\tau_N|\theta)$ and $\hat{g}_\rho(\tau_N|\theta)$.*

$$\| \mathbb{E}[\hat{g}_\rho(\tau_N|\theta)] - \nabla_\theta J_\rho(\theta)\| \leq \hat{\delta}_g \tag{26}$$

$$\mathbb{E} \|\hat{g}_\rho(\tau_N|\theta) - \nabla_\theta J_\rho(\theta)\| \leq \frac{\sqrt{d}G}{(1-\gamma)^2} + \hat{\delta}_g \tag{27}$$

$$\| \mathbb{E}[\hat{F}_\rho(\tau_N|\theta)] - F_\rho(\theta)\| \leq \hat{\delta}_F \tag{28}$$

$$\mathbb{E} \|\hat{F}_\rho(\tau_N|\theta) - F_\rho(\theta)\| \leq \sqrt{d}G^2 + \hat{\delta}_F \tag{29}$$

$$\text{Var}(\hat{g}_\rho(\tau_N|\theta)) \leq \frac{dG^2}{(1-\gamma)^4} \text{ and } \text{Var}(\hat{F}_\rho(\tau_N|\theta)) \leq dG^4 \tag{30}$$

*Where $\hat{\delta}_g = G\left(\frac{N+1}{1-\gamma} + \frac{\gamma}{(1-\gamma)^2}\right)\gamma^N$ and $\hat{\delta}_F = G^2\gamma^N$*

While classical algorithms produce unbiased estimators of $\nabla_\theta J_\rho(\theta)$ and $F_\rho(\theta)$ via sampling from geometric distributions, our deterministic approach in (14) and (15) utilizes truncation to allow integration into quantum systems. The proof of Theorem 1 (in Appendix C) analyzes the truncation-based estimators by comparing them to their following infinite-horizon counterpart as an intermediate step:

$$g(\tau|\theta) = \sum_{n=0}^{\infty} \left( \sum_{t=0}^{n} \nabla_\theta \log \pi_\theta(a_t|s_t) \right) (\gamma^n r(s_n, a_n))$$

$$F(\tau|\theta) = (1-\gamma) \sum_{n=0}^{\infty} \left( \gamma^n \nabla_\theta \log \pi_\theta(a_n|s_n) \otimes \log \pi_\theta(a_n|s_n) \right)$$

Analyzing the error bounds $\| \mathbb{E}[\hat{g}_\rho(\tau_N|\theta)] - \mathbb{E}[g(\tau|\theta)]\|$ and $\| \mathbb{E}[\hat{F}_\rho(\tau_N|\theta)] - \mathbb{E}[F(\tau|\theta)]\|$ quantifies the truncation error, showing that the truncation introduces an exponentially decaying bias of order $\mathcal{O}(\gamma^N)$ while maintaining a bounded variance. This is achieved by leveraging properties of the geometric series and analyzing the dependence on the discount factor $\gamma$. Given the above, we can formally state the bias and variance bounds for the gradient estimator $\tilde{g}_h$ and Fisher estimator $\tilde{F}_h$ for each $h$, along with expected number of quantum samples $|\psi_F^\theta\rangle$ and $|\psi_g^\theta\rangle$ used in (18) and (19).

**Theorem 2.** *For any $\theta$ and $h$, (18) and (19) satisfy $\mathbb{E}[\tilde{g}_h] = \mathbb{E}[\hat{g}_\rho(\tau_N|\theta)]$ and $\mathbb{E}[\tilde{F}_h] = \mathbb{E}[\hat{F}_\rho(\tau_N|\theta)]$ with reduced variance $\text{Var}(\tilde{g}_h) = \hat{\sigma}_g^2$ and $\text{Var}(\tilde{F}_h) = \hat{\sigma}_F^2$. Moreover, (18) and (19) perform $\tilde{\mathcal{O}}\left(\frac{G^2 d}{\hat{\sigma}_F}\right)$ queries of $\mathcal{U}_F$ and $\tilde{\mathcal{O}}\left(\frac{Gd}{(1-\gamma)^2\hat{\sigma}_g}\right)$ queries of $\mathcal{U}_g$ in expectations.*

The proof of Theorem 2 (in Appendix B) leverages Lemma 1 to achieve variance reduction by applying quantum mean estimation techniques to the truncated estimators, ensuring unbiased estimates of the policy gradient and Fisher matrix with reduced variance. Note that although $\tilde{g}_h$ and $\tilde{g}_h$ are unbiased with respect to $\hat{g}_\rho(\tau_N|\theta)$ and $\hat{F}_\rho(\tau_N|\theta)$, they are still biased with respect to the actual policy gradient and Fisher information matrix. Theorem 2 demonstrates that quantum variance reduction methods achieve a quadratic speedup in complexity compared to classical methods such as averaging, when reducing to the same variance. This step is crucial to achieve the overall polynomial speedup in the final result. We thus are now able to describe the first and second order errors for the quantum estimators $\tilde{F}_h$ and $\tilde{g}_h$.

**Lemma 5.** *Under assumption 1, for a fixed $\theta$, we have the following results on $\tilde{F}_h$ and $\tilde{g}_h$.*

$$\| \mathbb{E}[\tilde{g}_h] - \nabla_\theta J_\rho(\theta)\|^2 \leq \hat{\delta}_g^2 \tag{31}$$

$$\mathbb{E} \left[ \|\tilde{g}_h - \nabla_\theta J_\rho(\theta)\|^2 \right] \leq \hat{\sigma}_g^2 + \hat{\delta}_g^2 \tag{32}$$

$$|| \mathbb{E}[\tilde{F}_h] - F_\rho(\theta)||^2 \leq \hat{\delta}_F^2 \qquad (33)$$

$$\mathbb{E}\left[||\tilde{F}_h - F_\rho(\theta)||^2\right] \leq \hat{\sigma}_F^2 + \hat{\delta}_F^2 \qquad (34)$$

*Where $\hat{\delta}_g^2 = G^2(\frac{N+1}{1-\gamma} + \frac{\gamma}{(1-\gamma)^2})^2 \gamma^{2N}$ and $\hat{\delta}_F^2 = G^4 \gamma^{2N}$*

Note that $\hat{\delta}_g^2$ and $\hat{\delta}_F^2$ in Lemma 5 characterizes the exponentially decaying bias by truncation. The proof of Lemma 5 (in Appendix D) decomposes the second order error into the sum of bias and variance terms, which are futher bounded by Theorem 1 and Theorem 2 respectively. Compared with the classical NPG algorithms in (Mondal & Aggarwal, 2024a; Liu et al., 2020), the extra bias terms requires a more careful analysis. We derive the error bound for the first-order and second-order approximation of $\omega_k^*$ as follows:

**Lemma 6.** *Consider the NPG-finding loop* (21) *with $\alpha \leq \frac{\mu_F}{56G^4}$ and $G^2 \gamma^N \leq \frac{\mu_F}{8}$, for a fixed $\theta_k$, if assumptions 1-3 hold,*

$$\mathbb{E}\left[\|\omega_k - \omega_k^*\|^2\right] \leq \exp\left(-H\alpha\mu_F\right) \mathbb{E}\|\omega_0 - \omega^*\|^2 + C_0 \qquad (35)$$

*Additionally, we also have*

$$\|\mathbb{E}[\omega_k] - \omega_k^*\|^2 \leq \exp\left(-H\alpha\mu_F\right) \|\mathbb{E}[\omega_0] - \omega^*\|^2 + C_1 \qquad (36)$$

*where*

$$C_0 = 4\mu_F^{-2}\left[2\hat{\delta}_F^2 \frac{\mu_F^{-2}G^2}{(1-\gamma)^4} + \hat{\delta}_g^2\right]$$
$$+ 6\alpha\mu_F^{-1}\left[\frac{\mu_F^{-2}G^2}{(1-\gamma)^4}\left(\hat{\sigma}_F^2\hat{\delta}_F^2\right) + \left(\hat{\sigma}_g^2 + \hat{\delta}_g^2\right)\right] \qquad (37)$$
$$= \mathcal{O}\left(\gamma^{2N} + \frac{\hat{\sigma}_F^2}{(1-\gamma)^4} + \hat{\sigma}_g^2\right)$$

$$C_1 = \frac{6}{\mu_F}\left(\alpha + \frac{1}{\mu_F}\right)\left[\hat{\delta}_F^2\left\{\mathbb{E}\left[\|\omega_0 - \omega_k^*\|^2\right] + \alpha R_0 + R_1\right\} + \frac{\mu_F^{-2}G^2}{(1-\gamma)^4}\hat{\delta}_F^2 + \hat{\delta}_g^2\right] \qquad (38)$$
$$= \mathcal{O}\left(\gamma^{2N}\right)$$

*while $R_0 = 6\mu_F^{-1}\left[\frac{\mu_F^{-2}G^2}{(1-\gamma)^4}\left(\hat{\sigma}_F^2 + \hat{\delta}_F^2\right) + \left(\hat{\sigma}_g^2 + \hat{\delta}_g^2\right)\right]$, $R_1 = 4\mu_F^{-2}\left[2\hat{\delta}_F^2 \frac{\mu_F^{-2}G^2}{(1-\gamma)^4} + \hat{\delta}_g^2\right]$*

The proof of Lemma 6 (in Appendix E) proceeds by first expressing the error between the estimated gradient and the ideal gradient as a recursive update from the inner-loop iteration. It then demonstrates that each update contracts the error exponentially, meaning the error shrinks by a fixed multiplicative factor with each iteration. Subsequently, the proof carefully bounds the additional errors introduced by truncation bias and estimator variance, leading to explicit residual terms. Finally, these contraction and residual terms are combined through a recursive argument to establish the overall error guarantees.

Lemma 6 shows that the bias terms $\mathcal{O}(\gamma^N)$ from the gradient and Fisher matrix estimators would accumulate across iterations, leading to a total error contribution of the form

$$\mathcal{O}\left(\sum_{h=0}^{H-1} \gamma^{2N}\right) \approx \mathcal{O}\left(\frac{\gamma^{2N}}{1 - e^{-H\alpha\mu_F}}\right)$$

while the learning rate $\alpha$ is chosen appropriately to ensure exponential decay, eventually leading to the terms $C_0$ and $C_1$. We next combine the *outer loop* and *inner loop* analysis with the quantum speedup results to form the final result.

### 4.3. Final Result

Combining Lemma 6 and the fact that the last term in (25) is $\mathcal{O}(1/K)$. Thus, by taking $H = \mathcal{O}(\log(\epsilon^{-1}))$, $N = \mathcal{O}(\log(\epsilon^{-1}))$, $K = \mathcal{O}(\epsilon^{-1})$, $\hat{\sigma}_g^2 = \mathcal{O}(\epsilon)$ and $\hat{\sigma}_F^2 = \mathcal{O}((1-\gamma)^4\epsilon)$, we guarantee the optimality gap to be at most $\sqrt{\epsilon_{\text{bias}}} + \epsilon$. By Theorem 2, $\tilde{\mathcal{O}}\left(\frac{G^2 d}{(1-\gamma)^2\sqrt{\epsilon}}\right)$ queries of $\mathcal{U}_F$ and $\tilde{\mathcal{O}}\left(\frac{Gd}{(1-\gamma)^2\sqrt{\epsilon}}\right)$ queries of $\mathcal{U}_g$. This results in a sample complexity of $\tilde{\mathcal{O}}\left(HKN \cdot \frac{d}{(1-\gamma)^2\sqrt{\epsilon}}\right) = \tilde{\mathcal{O}}(1/\epsilon^{1.5})$ and an iteration complexity of $\mathcal{O}(K) = \tilde{\mathcal{O}}(1/\epsilon)$. The result is formalized in the following theorem.

**Theorem 3.** *Let $\{\theta_k\}_{k=0}^{K-1}$ be the policy parameters generated by Algorithm 1, $\pi^*$ be the optimal policy and $J_\rho^*$ denote the optimal value of $J_\rho(\cdot)$ corresponding to an initial distribution $\rho$. Assume that assumptions $1 - 3$ hold and the learning parameters are chosen as stated in Lemma 3 and Lemma 6. For all sufficiently small $\epsilon$, and $H = \mathcal{O}(\log(\epsilon^{-1}))$, $N = \mathcal{O}(\log(\epsilon^{-1}))$, $K = \mathcal{O}(\epsilon^{-1})$, $\hat{\sigma}_g^2 = \mathcal{O}(\epsilon)$ and $\hat{\sigma}_F^2 = \mathcal{O}((1-\gamma)^4\epsilon)$, the following holds.*

$$J_\rho^* - \frac{1}{K}\sum_{k=0}^{K-1} \mathbb{E}[J_\rho(\theta_k)] \leq \sqrt{\epsilon_{\text{bias}}} + \epsilon \qquad (39)$$

*This results in $\tilde{\mathcal{O}}(\epsilon^{-1.5})$ sample complexity and $\mathcal{O}(\epsilon^{-1})$ iteration complexity.*

The resulting sample complexity $\tilde{\mathcal{O}}(\epsilon^{-1.5})$ is due to the quantum speedup in the *inner loop* while the *outer loop* maintained as in the classical settings. This aligns with the results of quantum stochastic optimizations (Sidford & Zhang, 2024) due to the same double-loop algorithm structure with classical *outer loop* and quantum *inner loop*. Note that $H$ is chosen to be logarithmically dependent on the total samples to account for the exponentially decaying bias term of the truncation error.

## 5. Conclusion

This paper introduces the Quantum Natural Policy Gradient (QNPG) algorithm, which utilizes quantum mean estimation and a deterministic gradient sampling approach to address

sample efficiency challenges in infinite-horizon reinforcement learning. By adapting the sampling method in the classical state-of-the-art result to a quantum-compatible deterministic procedure, we achieve a sample complexity of $\tilde{\mathcal{O}}(\epsilon^{-1.5})$, improving upon the $\tilde{\mathcal{O}}(\epsilon^{-2})$ result of the classical state-of-the-art result. The algorithm balances the trade-off between deterministic encoding and bounded bias, demonstrating the feasibility of leveraging quantum acceleration in policy optimization.

While our study focuses on policy gradient methods, quantum actor–critic algorithms remain largely unexplored. Recent classical results offer order-optimal sample complexity guarantees (Gaur et al., 2024; Ganesh et al., 2025), making the search for quantum speedups in this setting a key open direction. Another open direction is extending the constrained setup studied in (Mondal & Aggarwal, 2024b) to the quantum setting.

## Acknowledgment

This work is supported in part by the National Center for Transportation Cybersecurity and Resiliency (TraCR) (a U.S. Department of Transportation National University Transportation Center) headquartered at Clemson University, Clemson, South Carolina, USA. Any opinions, findings, conclusions, and recommendations expressed in this material are those of the author(s) and do not necessarily reflect the views of TraCR, and the U.S. Government assumes no liability for the contents or use thereof.

## Impact Statement

This paper presents work whose goal is to advance the field of Machine Learning. There are many potential societal consequences of our work, none which we feel must be specifically highlighted here.

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

# A. Proof of Lemma 4

## A.1. Quantum Superpositioned Trajectories

We first explain the construction of the required unitary $\mathcal{U}_{P(\tau_N)}$ step by step, showing it uses one call to the initial-state oracle $\mathcal{U}_\rho$ and $\mathcal{O}(N)$ calls to the transition oracle $\mathcal{U}_P$ and policy oracle $\Pi$.

**Initial registers.** We start with the following (quantum) registers:

- A register $|\theta\rangle$ containing the policy parameters (which will remain untouched except as a control for $\Pi$).

- $N$ state registers $|0\rangle_{\mathsf{S}_0}, \ldots, |0\rangle_{\mathsf{S}_{N-1}}$.

- $N$ action registers $|0\rangle_{\mathsf{A}_0}, \ldots, |0\rangle_{\mathsf{A}_{N-1}}$.

Thus the total initial state is

$$|\theta\rangle \otimes |0\rangle^{\otimes 2N}.$$

**Load the initial state.** We apply $\mathcal{U}_\rho$ (see (5)) to the first state register $\mathsf{S}_0$, which transforms

$$|0\rangle_{\mathsf{S}_0} \xmapsto{\mathcal{U}_\rho} \sum_{s_0 \in \mathcal{S}} \sqrt{\rho(s_0)} \, |s_0\rangle_{\mathsf{S}_0}.$$

The other registers remain in the all-zero state. Therefore, the global state becomes

$$|\theta\rangle \otimes \sum_{s_0 \in \mathcal{S}} \sqrt{\rho(s_0)} \, |s_0\rangle_{\mathsf{S}_0} \otimes |0\rangle_{\mathsf{S}_1} \cdots |0\rangle_{\mathsf{S}_{N-1}} \otimes |0\rangle_{\mathsf{A}_0} \cdots |0\rangle_{\mathsf{A}_{N-1}}.$$

**Iterate over $t = 0, \ldots, N-1$.** For each time step $t$, we perform two sub-steps:

1. *Policy oracle $\Pi$.* By definition (see (6)), $\Pi$ implements

$$\Pi : \quad |\theta\rangle \, |s\rangle \, |0\rangle \quad \longmapsto \quad |\theta\rangle \, |s\rangle \sum_{a \in \mathcal{A}} \sqrt{\pi_\theta(a|s)} \, |a\rangle.$$

   We apply $\Pi$ to $|\theta\rangle$, the state register $\mathsf{S}_t$, and the action register $\mathsf{A}_t$. This sends

$$|s_t\rangle_{\mathsf{S}_t} \, |0\rangle_{\mathsf{A}_t} \quad \longmapsto \quad |s_t\rangle_{\mathsf{S}_t} \sum_{a_t \in \mathcal{A}} \sqrt{\pi_\theta(a_t|s_t)} \, |a_t\rangle_{\mathsf{A}_t}.$$

2. *Transition oracle $\mathcal{U}_P$.* By definition (see (4)), $\mathcal{U}_P$ implements

$$\mathcal{U}_P : \quad |s_t\rangle \, |a_t\rangle \, |0\rangle \quad \longmapsto \quad |s_t\rangle \, |a_t\rangle \sum_{s_{t+1} \in \mathcal{S}} \sqrt{P(s_{t+1}|s_t, a_t)} \, |s_{t+1}\rangle.$$

   We apply this to registers $\mathsf{S}_t$, $\mathsf{A}_t$, and $\mathsf{S}_{t+1}$, which transforms

$$|s_t, a_t, 0\rangle \quad \longmapsto \quad |s_t, a_t\rangle \sum_{s_{t+1} \in \mathcal{S}} \sqrt{P(s_{t+1}|s_t, a_t)} \, |s_{t+1}\rangle.$$

After applying $\Pi$ and $\mathcal{U}_P$ at each step $t$, the amplitudes multiply to give

$$\sqrt{\pi_\theta(a_t|s_t)} \times \sqrt{P\big(s_{t+1}|s_t, a_t\big)}.$$

**Final state.** After these $N$ steps, the global state is

$$|\theta\rangle \sum_{s_0, a_0, \ldots, s_{N-1}, a_{N-1}} \sqrt{\rho(s_0)} \prod_{t=0}^{N-1} \sqrt{\pi_\theta(a_t|s_t)\, P(s_{t+1}|s_t, a_t)} \,\big| s_0, a_0,\, s_1, a_1,\, \ldots,\, s_{N-1}, a_{N-1}\big\rangle.$$

Observe that the squared amplitude of each trajectory $\tau_N$ matches exactly $P_\theta(\tau_N)$. Hence,

$$\mathcal{U}_{P(\tau_N)}: \quad |\theta\rangle\, |0\rangle^{\otimes 2N} \quad \longmapsto \quad |\theta\rangle \sum_{\tau_N} \sqrt{P_\theta(\tau_N)}\,|\tau_N\rangle.$$

**Query complexity.**

- We call $\mathcal{U}_\rho$ *once* to load $s_0$ in the register $\mathsf{S}_0$.

- For each step $t = 0, \ldots, N-1$, we call $\Pi$ *once* and $\mathcal{U}_P$ *once*.

Thus, the total number of queries for one sample of $|\Psi_\theta\rangle$ to the "environment + policy" oracles is $1 + 2N = \mathcal{O}(N)$.

### A.2. Quantum NPG Estimators

**1. Constructing the superposition $|\Psi_\theta\rangle$.** By A.1, we have a unitary

$$\mathcal{U}_{P(\tau_N)} : |\theta\rangle\, |0\rangle^{\otimes 2N} \quad \longmapsto \quad |\theta\rangle \sum_{\tau_N} \sqrt{P_\theta(\tau_N)}\,|\tau_N\rangle,$$

implemented via one call to $\mathcal{U}_\rho$ and $\mathcal{O}(N)$ queries to the transition oracle $\mathcal{U}_P$ and the policy oracle $\Pi$. Hence we can prepare

$$|\Psi_\theta\rangle \;=\; |\theta\rangle \sum_{\tau_N} \sqrt{P_\theta(\tau_N)}\,|\tau_N\rangle,$$

a coherent superposition over all length-$N$ trajectories $\tau_N$ with amplitudes $\sqrt{P_\theta(\tau_N)}$.

**2. Defining the estimators $\hat{F}_\rho$ and $\hat{g}_\rho$.** We recall the definitions in (14) and (15):

$$\hat{g}_\rho(\tau_N|\theta) \;=\; \sum_{n=0}^{N-1} \left(\sum_{t=0}^{n} \nabla_\theta \log \pi_\theta(a_t|s_t)\right)\big(\gamma^n\, r(s_n, a_n)\big),$$

$$\hat{F}_\rho(\tau_N|\theta) \;=\; (1-\gamma) \sum_{n=0}^{N-1} \gamma^n \Big(\nabla_\theta \log \pi_\theta(a_n|s_n)\Big) \otimes \Big(\nabla_\theta \log \pi_\theta(a_n|s_n)\Big).$$

Because $\pi_\theta(\cdot|\cdot)$, $r(\cdot, \cdot)$, and discounting can be computed classically in polynomial time, these functions are straightforwardly computable given full knowledge of $\tau_N$.

**3. Quantum evaluation of classical functions.** A standard result in quantum computing states that any efficiently computable function $f(x)$ admits a polynomial-size quantum circuit implementing

$$|x\rangle\, |0\rangle \quad \longmapsto \quad |x\rangle\, |f(x)\rangle$$

in coherence. Applying this principle:

- Let $\mathcal{U}_F$ be the unitary that computes $\hat{F}_\rho(\tau_N|\theta)$ given $\tau_N$. Concretely:

$$\mathcal{U}_F : |\tau_N\rangle\, |0\rangle \quad \longmapsto \quad |\tau_N\rangle\, |\hat{F}_\rho(\tau_N|\theta)\rangle.$$

- Let $\mathcal{U}_g$ be the unitary that computes $\hat{g}_\rho(\tau_N \mid \theta)$ similarly.

These circuits require no additional calls to the environment oracles, since they merely read off the classical data $(s_t, a_t)$ in the registers and perform the classical computations needed for $\nabla_\theta \log \pi_\theta$, discounting, and rewards.

**4. Acting on the superposition $|\Psi_\theta\rangle$.** Once we have $|\Psi_\theta\rangle = |\theta\rangle \sum_{\tau_N} \sqrt{P_\theta(\tau_N)} |\tau_N\rangle$, we append an auxiliary register $|0\rangle$ and apply $\mathcal{U}_F$ or $\mathcal{U}_g$ in superposition. By linearity:

$$\big(\mathrm{Id} \otimes \mathcal{U}_F\big)\Big(|\Psi_\theta\rangle \otimes |0\rangle\Big) \;=\; |\theta\rangle \sum_{\tau_N} \sqrt{P_\theta(\tau_N)} |\tau_N\rangle \;\otimes\; |\hat{F}_\rho(\tau_N|\theta)\rangle.$$

Analogously for $\mathcal{U}_g$. This precisely shows

$$\mathcal{U}_F\Big(|\Psi_\theta\rangle \otimes |0\rangle\Big) \;=\; |\theta\rangle \sum_{\tau_N} \sqrt{P_\theta(\tau_N)} |\tau_N\rangle \otimes |\hat{F}_\rho(\tau_N|\theta)\rangle,$$

and similarly for $\hat{g}_\rho(\tau_N|\theta)$.

**5. Query complexity.**

- *Preparing the superposition $|\Psi_\theta\rangle$:* This needs $\mathcal{O}(N)$ queries to $\mathcal{U}_P$ (the transition oracle) and $\Pi$ (the policy oracle), plus one query to $\mathcal{U}_\rho$.

- *Computing $\hat{F}_\rho$ or $\hat{g}_\rho$ in superposition:* The unitaries $\mathcal{U}_F$ and $\mathcal{U}_g$ are classical function simulations that do not require extra queries to $\mathcal{U}_P$. They simply act on the $(s_t, a_t)$ registers directly.

Hence, the overall cost in environment queries remains $\mathcal{O}(N)$ for a length-$N$ trajectory.

## B. Details of Quantum Variance Reduction

---

**Algorithm 2** `QVarianceReduce` (Algorithm 2 in (Sidford & Zhang, 2024))

---

1: **Input:** Random variable $X$, target variance $\hat{\sigma}^2$, `QuantumMeanEstimation`$^+$ from Algorithm 3
2: **Output:** An unbiased estimate $\hat{\mu}$ of $X$ with variance at most $\hat{\sigma}^2$
3: Set $\tilde{\mu}_0 \leftarrow$ `QuantumMeanEstimation`$^+(X, \hat{\sigma}/10)$
4: Randomly sample $j \sim \mathrm{Geom}\left(\frac{1}{2}\right) \in \mathbb{N}$
5: $\tilde{\mu}_j \leftarrow$ `QuantumMeanEstimation`$^+(X, 2^{-3j/4}\hat{\sigma}/10)$
6: $\tilde{\mu}_{j-1} \leftarrow$ `QuantumMeanEstimation`$^+(X, 2^{-3(j-1)/4}\hat{\sigma}/10)$
7: $\hat{\mu} \leftarrow \tilde{\mu}_0 + 2^j(\tilde{\mu}_j - \tilde{\mu}_{j-1})$
8: **Return:** $\hat{\mu}$

---

---

**Algorithm 3** `QuantumMeanEstimation`$^+$ (Algorithm 1 in (Sidford & Zhang, 2024))

---

1: **Input:** Random variable $X$, target variance $\hat{\sigma}^2 \leq L^2$, `QuantumMeanEstimation` from Lemma 1, parameters $\delta = \hat{\sigma}^6/(4L)^6$ and $D = \frac{\hat{\sigma}}{4} + \frac{16L^3}{\hat{\sigma}^2}$
2: **Output:** An estimate $\hat{\mu}$ satisfying $\mathbb{E}\left\|\hat{\mu} - \mathbb{E}[X]\right\|^2 \leq \hat{\sigma}^2$
3: Set $X_1 \leftarrow$ `QuantumMeanEstimation`$(X, \hat{\sigma}/4, \delta)$
4: Randomly draw a classical sample $X_2$ of $X$
5: **if** $\|X_1 - X_2\| \leq D$ **then**
6:     **Return:** $X_1$
7: **else**
8:     Randomly draw one classical sample $X_3$ of $X$
9:     **Return:** $X_3$
10: **end if**

---

Algorithm 2 is the detailed description of `QVarianceReduce`, which provides a quadratic sample complexity speedup shown in the following Lemma:

**Lemma 7** (Theorem 4 of (Sidford & Zhang, 2024))**.** *Algorithm 2 returns the desired result using expected $\tilde{\mathcal{O}}(L\sqrt{d}/\hat{\sigma})$ queries, with random variable $X$ having dimension $d$ and variance bounded by $L^2$.*

### B.1. Proof of Theorem 2

This is a direct result of combining Lemma 7 with the variance bounds of $\hat{g}_\rho(\tau_N|\theta)$ and $\hat{F}_\rho(\tau_N|\theta)$ in (30).

## C. Proof of Theorem 1

For a fixed $\theta$, we first note that $g(\tau|\theta)$ and $F(\tau|\theta)$ constructed as below would be an unbiaed estimator for $\nabla_\theta J_\rho(\theta)$ and $F_\rho(\theta)$:

$$g(\tau|\theta) = \sum_{n=0}^{\infty} \left( \sum_{t=0}^{n} \nabla_\theta \log \pi_\theta(a_t|s_t) \right) (\gamma^n r(s_n, a_n)), \tag{40}$$

$$F(\tau|\theta) = (1-\gamma) \sum_{n=0}^{\infty} \left( \gamma^n \nabla_\theta \log \pi_\theta(a_n|s_n) \otimes \log \pi_\theta(a_n|s_n) \right), \tag{41}$$

where $\tau = (s_0, a_0, \ldots, s_k, a_k, \ldots)$ is an infinite length trajectory, generated by taking policy $\pi_\theta$ with $s_0$ sampled from $\rho$. (40) is a standard expression of the policy gradient unbiased estimator used in (Baxter & Bartlett, 2001; Liu et al., 2020). To prove that (41) is an unbiased estimator of $F_\rho(\theta)$, note that

$$\mathbb{E}_\tau[\hat{F}(\tau|\theta)] = \mathbb{E}_{\tau \sim (\rho, \pi_\theta)} \left[ (1-\gamma) \sum_{n=0}^{\infty} \gamma^n \nabla_\theta \log \pi_\theta(a_n|s_n) \otimes \nabla_\theta \log \pi_\theta(a_n|s_n) \right] \tag{42}$$

$$= (1-\gamma) \sum_{n=0}^{\infty} \gamma^n \mathbb{E}_\tau \left[ \nabla_\theta \log \pi_\theta(a_n|s_n) \otimes \nabla_\theta \log \pi_\theta(a_n|s_n) \right] \tag{43}$$

$$= (1-\gamma) \sum_{n=0}^{\infty} \gamma^n \sum_{s,a} \Pr(s_n = s) \pi_\theta(a|s) \nabla_\theta \log \pi_\theta(a|s) \otimes \nabla_\theta \log \pi_\theta(a|s) \tag{44}$$

$$= \sum_{s,a} \left[ (1-\gamma) \sum_{n=0}^{\infty} \gamma^n \Pr(s_n = s) \right] \pi_\theta(a|s) \nabla_\theta \log \pi_\theta(a|s) \otimes \nabla_\theta \log \pi_\theta(a|s) \tag{45}$$

$$= \sum_{s,a} d_\rho^\pi(s) \pi_\theta(a|s) \nabla_\theta \log \pi_\theta(a|s) \otimes \nabla_\theta \log \pi_\theta(a|s) \tag{46}$$

$$= \sum_{s,a} \nu_\rho^\pi(s, a) \nabla_\theta \log \pi_\theta(a|s) \otimes \nabla_\theta \log \pi_\theta(a|s) \tag{47}$$

$$= F_\rho(\theta). \tag{48}$$

Therefore we have

$$|| \mathbb{E}[\hat{g}_\rho(\tau_N|\theta)] - \nabla_\theta J_\rho(\theta)|| = || \mathbb{E}[\hat{g}_\rho(\tau_N|\theta)] - \mathbb{E}[g(\tau|\theta)]||$$

$$= \left\| \mathbb{E}\left[ \sum_{n=N}^{\infty} \left( \sum_{t=0}^{n} \nabla_\theta \log \pi_\theta(a_t|s_t) \right) (\gamma^n r(s_n, a_n)) \right] \right\|$$

$$\leq || G \sum_{h=N}^{\infty} (n+1)\gamma^n ||$$

$$= G \left( \frac{N+1}{1-\gamma} + \frac{\gamma}{(1-\gamma)^2} \right) \gamma^N \tag{49}$$

Furthermore, we have

$$\mathbb{E} ||\hat{g}_\rho(\tau_N|\theta) - \nabla_\theta J_\rho(\theta)|| \leq \mathbb{E} ||\hat{g}_\rho(\tau_N|\theta) - \mathbb{E}[\hat{g}_\rho(\tau_N|\theta)]|| + \mathbb{E} || \mathbb{E}[\hat{g}_\rho(\tau_N|\theta)] - \nabla_\theta J_\rho(\theta)|| \tag{50}$$

Since we also have $||\hat{g}_\rho(\tau_N|\theta)|| \leq \frac{G}{(1-\gamma)^2}$ from Lemma 2, we further have

$$\text{Var}(\hat{g}_\rho(\tau_N|\theta)) = \mathbb{E} ||\hat{g}_\rho(\tau_N|\theta) - \mathbb{E}[\hat{g}_\rho(\tau_N|\theta)]||^2 \leq \frac{dG^2}{(1-\gamma)^4} \tag{51}$$

Since $\mathbb{E}\,||\hat{g}_\rho(\tau_N|\theta) - \mathbb{E}[\hat{g}_\rho(\tau_N|\theta)]|| \leq \sqrt{\mathbb{E}\,||\hat{g}_\rho(\tau_N|\theta) - \mathbb{E}[\hat{g}_\rho(\tau_N|\theta)]||^2}$ due to Jensen's inequality, combining (49), (50) and (51) we obtain (27). Similarly,

$$
\begin{aligned}
||\,\mathbb{E}[\hat{F}_\rho(\tau_N|\theta)] - F_\rho(\theta)|| &= ||\,\mathbb{E}[\hat{F}_\rho(\tau_N|\theta)] - \mathbb{E}[F(\tau|\theta)]|| \\
&= \left|\left|\,\mathbb{E}\left[(1-\gamma)\sum_{n=N}^{\infty}(\nabla_\theta \log \pi_\theta(a_n|s_n) \otimes \nabla_\theta \log \pi_\theta(a_n|s_n)\gamma^n)\right]\right|\right| \\
&\leq G^2\gamma^N
\end{aligned}
\tag{52}
$$

Furthermore, we have

$$
\mathbb{E}\,||\hat{F}_\rho(\tau_N|\theta) - F_\rho(\theta)|| \leq \mathbb{E}\,||\hat{F}_\rho(\tau_N|\theta) - \mathbb{E}[\hat{F}_\rho(\tau_N|\theta)]|| + \mathbb{E}\,||\,\mathbb{E}[\hat{F}_\rho(\tau_N|\theta)] - F_\rho(\theta)||
\tag{53}
$$

Since $||\hat{F}_\rho(\tau_N|\theta)|| \leq G^2$, which is by the definition of $\hat{F}_\rho(\tau_N|\theta)$, we further have

$$
\mathrm{Var}(\hat{F}_\rho(\tau_N|\theta)) = \mathbb{E}\,||\hat{F}_\rho(\tau_N|\theta) - \mathbb{E}[\hat{F}_\rho(\tau_N|\theta)]||^2 \leq dG^4
\tag{54}
$$

Since $\mathbb{E}\,||\hat{F}_\rho(\tau_N|\theta) - \mathbb{E}[\hat{F}_\rho(\tau_N|\theta)]|| \leq \sqrt{\mathbb{E}\,||\hat{F}_\rho(\tau_N|\theta) - \mathbb{E}[\hat{g}_\rho(\tau_N|\theta)]||^2}$ due to Jensen's inequality, combining (52), (53) and (54) we obtain (29).

## D. Proof of Lemma 5

Since Algorithm 2 outputs an unbiased estimate of the input, we have that for any $\theta$, $\mathbb{E}[\hat{F}_\rho(\tau_N|\theta)] = E[\tilde{g}_h]$ and $\mathbb{E}[\hat{F}_\rho(\tau_N|\theta)] = E[\tilde{F}_h]$. Thus, (31) and (33) directly follow from (26) and (28). For (32) and (34), we note that

$$
\mathbb{E}\left[||\tilde{g}_h - \nabla_\theta J_\rho(\theta)||^2\right] = \mathbb{E}\left[||\tilde{g}_h - \mathbb{E}(\tilde{g}_h) + \mathbb{E}(\tilde{g}_h) - \nabla_\theta J_\rho(\theta)||^2\right] = \mathbb{E}\left[||\tilde{g}_h - \mathbb{E}(\tilde{g}_h)||^2\right] + ||\,\mathbb{E}(\tilde{g}_h) - \nabla_\theta J_\rho(\theta)||^2
\tag{55}
$$

since the cross term contain $\mathbb{E}[\tilde{g}_h - \mathbb{E}(\tilde{g}_h)]$, which equals to zero. We further have

$$
||\,\mathbb{E}(\tilde{g}_h) - \nabla_\theta J_\rho(\theta)||^2 \leq G^2\left(\frac{N+1}{1-\gamma} + \frac{\gamma}{(1-\gamma)^2}\right)^2\gamma^{2N} \quad \text{and} \quad \mathbb{E}\left[||\tilde{g}_h - \mathbb{E}(\tilde{g}_h)||^2\right] \leq \hat{\sigma}_g^2
\tag{56}
$$

Combining the above two equation we obtain (32). (34) can be derived by the same procedure. We have

$$
\mathbb{E}\left[||\tilde{F}_h - F_\rho(\theta)||^2\right] = \mathbb{E}\left[||\tilde{F}_h - \mathbb{E}(\tilde{F}_h) + \mathbb{E}(\tilde{F}_h) - F_\rho(\theta)||^2\right] = \mathbb{E}\left[||\tilde{F}_h - \mathbb{E}(\tilde{F}_h)||^2\right] + ||\,\mathbb{E}(\tilde{F}_h) - F_\rho(\theta)||^2
\tag{57}
$$

since the cross term contain $\mathbb{E}[\tilde{F}_h - \mathbb{E}(\tilde{F}_h)]$, which equals to zero. We further have

$$
||\,\mathbb{E}(\tilde{F}_h) - F_\rho(\theta)||^2 \leq \frac{G^4\gamma^{2N}}{(1-\gamma)^2} \quad \text{and} \quad \mathbb{E}\left[||\tilde{F}_h - \mathbb{E}(\tilde{F}_h)||^2\right] \leq \hat{\sigma}_F^2
\tag{58}
$$

Combining the above two equation we obtain (34).

## E. Proof of Lemma 6

Throughout the proof, we assume a fixed $k^{th}$ iteration of outer loop with the fixed policy parameter $\theta_k$. Let $\nabla_\omega \tilde{L}_h = \nabla_\omega \tilde{L}(\omega, \tau_N)\big|_{\omega=\omega_h} = \tilde{F}_h\omega_h - \tilde{g}_h$. To prove the first statement, observe the following relations.

$$
\begin{aligned}
\|\omega_{h+1} - \omega_k^*\|^2 &= \|\omega_h - \alpha\nabla_\omega\tilde{L}_h - \omega_k^*\|^2 \\
&= \|\omega_h - \omega_k^*\|^2 - 2\alpha\langle\omega_h - \omega_k^*, \nabla_\omega\tilde{L}_h\rangle + \alpha^2\|\nabla_\omega\tilde{L}_h\|^2 \\
&= \|\omega_h - \omega_k^*\|^2 - 2\alpha\langle\omega_h - \omega_k^*, F_\rho(\theta_k)(\omega_h - \omega_k^*)\rangle \\
&\quad - 2\alpha\langle\omega_h - \omega_k^*, \nabla_\omega\tilde{L}_h - F_\rho(\theta_k)(\omega_h - \omega_k^*)\rangle + \alpha^2\|\nabla_\omega\tilde{L}_h\|^2 \\
&\overset{(a)}{\leq} \|\omega_h - \omega_k^*\|^2 - 2\alpha\mu_F\|\omega_h - \omega_k^*\|^2 - 2\alpha\langle\omega_h - \omega_k^*, \nabla_\omega\tilde{L}_h - F_\rho(\theta_k)(\omega_h - \omega_k^*)\rangle \\
&\quad + 2\alpha^2\|\nabla_\omega\tilde{L}_h - F_\rho(\theta_k)(\omega_h - \omega_k^*)\|^2 + 2\alpha^2\|F_\rho(\theta_k)(\omega_h - \omega_k^*)\|^2 \\
&\overset{(b)}{\leq} \|\omega_h - \omega_k^*\|^2 - 2\alpha\mu_F\|\omega_h - \omega_k^*\|^2 - 2\alpha\langle\omega_h - \omega_k^*, \nabla_\omega\tilde{L}_h - F_\rho(\theta_k)(\omega_h - \omega_k^*)\rangle \\
&\quad + 2\alpha^2\|\nabla_\omega\tilde{L}_h - F_\rho(\theta_k)(\omega_h - \omega_k^*)\|^2 + 2G^4\alpha^2\|\omega_h - \omega_k^*\|^2
\end{aligned}
$$

where $(a)$ and $(b)$ follow from $\mu_F \leq \|F_\rho(\theta_k)\| \leq G^2$. Taking conditional expectation on both sides, we obtain

$$\mathbb{E}\left[\|\omega_{h+1} - \omega_k^*\|^2\right] \leq (1 - 2\alpha\mu_F + 2G^4\alpha^2)\|\omega_h - \omega_k^*\|^2 - 2\alpha\langle\omega_h - \omega_k^*, \mathbb{E}\left[\nabla_\omega \tilde{L}_h - F_\rho(\theta_k)(\omega_h - \omega_k^*)\right]\rangle$$
$$+ 2\alpha^2 \mathbb{E}\left\|\nabla_\omega \tilde{L}_h - F_\rho(\theta_k)(\omega_h - \omega_k^*)\right\|^2 \tag{59}$$

Observe that the third term in (59) can be bounded as follows.

$$\|\nabla_\omega \tilde{L}_h - F_\rho(\theta_k)(\omega_h - \omega_k^*)\|^2 = \|(\tilde{F}_h - F_\rho(\theta_k))(\omega_h - \omega_k^*) + (\tilde{F}_h - F_\rho(\theta_k))\omega_k^* + (\nabla_\theta J_\rho(\theta_k) - \tilde{g}_h)\|^2$$
$$\leq 3\|\tilde{F}_h - F_\rho(\theta_k)\|^2\|\omega_h - \omega_k^*\|^2 + 3\|\tilde{F}_h - F_\rho(\theta_k)\|^2\|\omega_k^*\|^2 + 3\|\nabla_\theta J_\rho(\theta_k) - \tilde{g}_h\|^2$$
$$\leq 3\|\tilde{F}_h - F_\rho(\theta_k)\|^2\|\omega_h - \omega_k^*\|^2 + \frac{3\mu_F^{-2}G^2}{(1-\gamma)^4}\|\tilde{F}_h - F_\rho(\theta_k)\|^2 + 3\|\nabla_\theta J_\rho(\theta_k) - \tilde{g}_h\|^2$$

where the last inequality follows from $\|\omega_k^*\|^2 = \|F_\rho(\theta_k)^{-1}\nabla_\theta J_\rho(\theta_k)\|^2 \leq \frac{\mu_F^{-2}G^2}{(1-\gamma)^4}$. Taking expectation yields

$$\mathbb{E}\|\nabla_\omega \tilde{L}_h - F_\rho(\theta_k)(\omega_h - \omega_k^*)\|^2$$
$$\leq 3\mathbb{E}\|\tilde{F}_h - F_\rho(\theta_k)\|^2\|\omega_h - \omega_k^*\|^2 + \frac{3\mu_F^{-2}G^2}{(1-\gamma)^4}\mathbb{E}\|\tilde{F}_h - F_\rho(\theta_k)\|^2 + 3\mathbb{E}\|\nabla_\theta J_\rho(\theta_k) - \tilde{g}_h\|^2$$
$$\leq 3\left[\hat{\sigma}_F^2 + \hat{\delta}_F^2\right]\|\omega_h - \omega_k^*\|^2 + \frac{3\mu_F^{-2}G^2}{(1-\gamma)^4}\left(\hat{\sigma}_F^2 + \hat{\delta}_F^2\right) + \left(3\hat{\sigma}_g^2 + 3\hat{\delta}_g^2\right) \tag{60}$$

The second term in (59) can be bounded as

$$-\langle\omega_h - \omega_k^*, \mathbb{E}[\nabla_\omega \tilde{L}_h - F_\rho(\theta_k)(\omega_h - \omega_k^*)]\rangle$$
$$\leq \frac{\mu_F}{4}\|\omega_h - \omega_k^*\|^2 + \frac{1}{\mu_F}\left\|\mathbb{E}[\nabla_\omega \tilde{L}_h - F_\rho(\theta_k)(\omega_h - \omega_k^*)]\right\|^2$$
$$\leq \frac{\mu_F}{4}\|\omega_h - \omega_k^*\|^2 + \frac{1}{\mu_F}\left\|\left\{\mathbb{E}[\tilde{F}_h] - F_\rho(\theta_k)\right\}\omega_h + \left\{\nabla_\theta J_\rho(\theta_k) - \mathbb{E}[\tilde{g}_h]\right\}\right\|^2$$
$$\leq \frac{\mu_F}{4}\|\omega_h - \omega_k^*\|^2 + \frac{\left(2\hat{\sigma}_F^2 + 2\hat{\delta}_F^2\right)\|\omega_h\|^2 + \left(2\hat{\sigma}_g^2 + 2\hat{\delta}_g^2\right)}{\mu_F}$$
$$\leq \frac{\mu_F}{4}\|\omega_h - \omega_k^*\|^2 + \frac{\left(4\hat{\sigma}_F^2 + 4\hat{\delta}_F^2\right)\|\omega_h - \omega_k^*\|^2 + \left(4\hat{\sigma}_F^2 + 4\hat{\delta}_F^2\right)\frac{\mu_F^{-2}G^2}{(1-\gamma)^4} + \left(2\hat{\sigma}_g^2 + 2\hat{\delta}_g^2\right)}{\mu_F} \tag{61}$$

where the last inequality follows from $\|\omega_k^*\|^2 = \|F_\rho(\theta_k)^{-1}\nabla_\theta J_\rho(\theta_k)\|^2 \leq \frac{\mu_F^{-2}G^2}{(1-\gamma)^4}$. Substituting the above bounds in (59), we have

$$\mathbb{E}\left[\|\omega_{h+1} - \omega_k^*\|^2\right] \leq \left(1 - \frac{3\alpha\mu_F}{2} + \frac{8\alpha\hat{\delta}_F^2}{\mu_F} + 6\alpha^2\left(\hat{\sigma}_F^2 + \hat{\delta}_F^2\right) + 2\alpha^2 G^4\right)\|\omega_h - \omega_k^*\|^2 + \frac{4\alpha}{\mu_F}\left[2\hat{\delta}_F^2\frac{\mu_F^{-2}G^2}{(1-\gamma)^4} + \hat{\delta}_g^2\right]$$
$$+ 6\alpha^2\left[\frac{\mu_F^{-2}G^2}{(1-\gamma)^4}\left(\hat{\sigma}_F^2 + \hat{\delta}_F^2\right) + \left(\hat{\sigma}_g^2 + \hat{\delta}_g^2\right)\right]$$

For $\hat{\delta}_F \leq \mu_F/8$, and $\alpha \leq \mu_F/[4(6\hat{\sigma}_F^2 + 6\hat{\delta}_F^2 + 2G^4)]$, we can modify the above inequality to the following.

$$\mathbb{E}[\|\omega_{h+1} - \omega_k^*\|^2] \leq (1 - \alpha\mu_F)\|\omega_h - \omega_k^*\|^2 + \frac{4\alpha}{\mu_F}\left[2\hat{\delta}_F^2\frac{\mu_F^{-2}G^2}{(1-\gamma)^4} + \hat{\delta}_g^2\right] + 6\alpha^2\left[\frac{\mu_F^{-2}G^2}{(1-\gamma)^4}\left(\hat{\sigma}_F^2 + \hat{\delta}_F^2\right) + \left(\hat{\sigma}_g^2 + \hat{\delta}_g^2\right)\right]$$

Taking expectation on both sides and unrolling the recursion yields

$$\mathbb{E}[\|\omega_H - \omega_k^*\|^2]$$

$$\leq (1 - \alpha\mu_F)^H \, \mathbb{E}\|\omega_0 - \omega_k^*\|^2 + \sum_{h=0}^{H-1} (1 - \alpha\mu_F)^h \left\{ \frac{4\alpha}{\mu_F} \left[ 2\hat{\delta}_F^2 \frac{\mu_F^{-2}G^2}{(1-\gamma)^4} + \hat{\delta}_g^2 \right] + 6\alpha^2 \left[ \frac{\mu_F^{-2}G^2}{(1-\gamma)^4} \left( \hat{\sigma}_F^2 + \hat{\delta}_F^2 \right) + \left( \hat{\sigma}_g^2 + \hat{\delta}_g^2 \right) \right] \right\}$$

$$\leq \exp\left(-H\alpha\mu_F\right) \mathbb{E}\|\omega_0 - \omega_k^*\|^2 + \frac{1}{\alpha\mu_F} \left\{ \frac{4\alpha}{\mu_F} \left[ 2\hat{\delta}_F^2 \frac{\mu_F^{-2}G^2}{(1-\gamma)^4} + \hat{\delta}_g^2 \right] + 6\alpha^2 \left[ \frac{\mu_F^{-2}G^2}{(1-\gamma)^4} \left( \hat{\sigma}_F^2 + \hat{\delta}_F^2 \right) + \left( \hat{\sigma}_g^2 + \hat{\delta}_g^2 \right) \right] \right\}$$

$$= \exp\left(-H\alpha\mu_F\right) \mathbb{E}\|\omega_0 - \omega_k^*\|^2 + \left\{ 4\mu_F^{-2} \left[ 2\hat{\delta}_F^2 \frac{\mu_F^{-2}G^2}{(1-\gamma)^4} + \hat{\delta}_g^2 \right] + 6\alpha\mu_F^{-1} \left[ \frac{\mu_F^{-2}G^2}{(1-\gamma)^4} \left( \hat{\sigma}_F^2 + \hat{\delta}_F^2 \right) + \left( \hat{\sigma}_g^2 + \hat{\delta}_g^2 \right) \right] \right\} \tag{62}$$

To prove the second statement, observe that we have the following recursion.

$$\| \mathbb{E}[\omega_{h+1}] - \omega_k^*\|^2 = \| \mathbb{E}[\omega_h] - \alpha \, \mathbb{E}[\nabla_\omega \tilde{L}_h] - \omega_k^*\|^2$$

$$= \| \mathbb{E}[\omega_h] - \omega_k^*\|^2 - 2\alpha\langle \mathbb{E}[\omega_h] - \omega_k^*, \mathbb{E}[g_h]\rangle + \alpha^2 \| \mathbb{E}[\nabla_\omega \tilde{L}_h]\|^2$$

$$= \| \mathbb{E}[\omega_h] - \omega_k^*\|^2 - 2\alpha\langle \mathbb{E}[\omega_h] - \omega_k^*, F_\rho(\theta_k)(\mathbb{E}[\omega_h] - \omega_k^*)\rangle$$

$$- 2\alpha\langle \mathbb{E}[\omega_h] - \omega_k^*, \mathbb{E}[\nabla_\omega \tilde{L}_h] - F_\rho(\theta_k)(\mathbb{E}[\omega_h] - \omega_k^*)\rangle + \alpha^2 \| \mathbb{E}[\nabla_\omega \tilde{L}_h]\|^2$$

$$\overset{(a)}{\leq} \| \mathbb{E}[\omega_h] - \omega_k^*\|^2 - 2\alpha\mu_F \| \mathbb{E}[\omega_h] - \omega_k^*\|^2 - 2\alpha\langle \mathbb{E}[\omega_h] - \omega_k^*, \mathbb{E}[\nabla_\omega \tilde{L}_h] - F_\rho(\theta_k)(\mathbb{E}[\omega_h] - \omega_k^*)\rangle$$

$$+ 2\alpha^2 \| \mathbb{E}[\nabla_\omega \tilde{L}_h] - F_\rho(\theta_k)(\mathbb{E}[\omega_h] - \omega_k^*)\|^2 + 2\alpha^2 \|F_\rho(\theta_k)(\mathbb{E}[\omega_h] - \omega_k^*)\|^2$$

$$\overset{(b)}{\leq} \| \mathbb{E}[\omega_h] - \omega_k^*\|^2 - 2\alpha\mu_F \| \mathbb{E}[\omega_h] - \omega_k^*\|^2 - 2\alpha\langle \mathbb{E}[\omega_h] - \omega_k^*, \mathbb{E}[\nabla_\omega \tilde{L}_h] - F_\rho(\theta_k)(\mathbb{E}[\omega_h] - \omega_k^*)\rangle$$

$$+ 2\alpha^2 \| \mathbb{E}[\nabla_\omega \tilde{L}_h] - F_\rho(\theta_k)(\mathbb{E}[\omega_h] - \omega_k^*)\|^2 + 2G^4\alpha^2 \| \mathbb{E}[\omega_h] - \omega_k^*\|^2$$

$$\leq (1 - 2\alpha\mu_F + 2G^4\alpha^2) \| \mathbb{E}[\omega_h] - \omega_k^*\|^2 - 2\alpha\langle \mathbb{E}[\omega_h] - \omega_k^*, \mathbb{E}[\nabla_\omega \tilde{L}_h] - F_\rho(\theta_k)(\mathbb{E}[\omega_h] - \omega_k^*)\rangle$$

$$+ 2\alpha^2 \left\| \mathbb{E}[\nabla_\omega \tilde{L}_h] - F_\rho(\theta_k)(\mathbb{E}[\omega_h] - \omega_k^*) \right\|^2 \tag{63}$$

where $(a)$ and $(b)$ follow from $\mu_F \leq \|F_\rho(\theta_k)\| \leq G^2$. The third term in the last line of (63) can be bounded as follows.

$$\| \mathbb{E}[\nabla_\omega \tilde{L}_h] - F_\rho(\theta_k)(\mathbb{E}[\omega_h] - \omega_k^*)\|^2$$

$$= \left\| \mathbb{E}\left[ (\tilde{F}_h - F_\rho(\theta_k))(\omega_h - \omega_k^*) \right] + (\mathbb{E}[\tilde{F}_h] - F_\rho(\theta_k))\omega_k^* + (\nabla_\theta J_\rho(\theta_k) - \mathbb{E}[\tilde{g}_h]) \right\|^2$$

$$\leq 3 \, \mathbb{E}\left[ \| \mathbb{E}[\tilde{F}_h] - F_\rho(\theta_k)\|^2 \|\omega_h - \omega_k^*\|^2 \right] + 3 \, \mathbb{E}\left[ \| \mathbb{E}[\tilde{F}_h] - F_\rho(\theta_k)\|^2 \right] \|\omega_k^*\|^2 + 3 \, \|\nabla_\theta J_\rho(\theta_k) - \mathbb{E}[\tilde{g}_h]\|^2$$

$$\leq 3\hat{\delta}_F^2 \, \mathbb{E}\left[ \|\omega_h - \omega_k^*\|^2 \right] + \frac{3\mu_F^{-2}G^2}{(1-\gamma)^4}\hat{\delta}_F^2 + 3\hat{\delta}_g^2$$

$$\overset{(a)}{\leq} 3\hat{\delta}_F^2 \left\{ \mathbb{E}\left[ \|\omega_0 - \omega_k^*\|^2 \right] + \alpha R_0 + R_1 \right\} + \frac{3\mu_F^{-2}G^2}{(1-\gamma)^4}\hat{\delta}_F^2 + 3\hat{\delta}_g^2$$

where $R_0 = 6\mu_F^{-1} \left[ \frac{\mu_F^{-2}G^2}{(1-\gamma)^4} \left( \hat{\sigma}_F^2 + \hat{\delta}_F^2 \right) + \left( \hat{\sigma}_g^2 + \hat{\delta}_g^2 \right) \right]$, $R_1 = 4\mu_F^{-2} \left[ 2\hat{\delta}_F^2 \frac{\mu_F^{-2}G^2}{(1-\gamma)^4} + \hat{\delta}_g^2 \right]$ and $(a)$ follows from (62). The second term in the last line of (63) can be bounded as follows.

$$-\langle \mathbb{E}[\omega_h] - \omega_k^*, \mathbb{E}\left[ \mathbb{E}[\nabla_\omega \tilde{L}_h] - F_\rho(\theta_k)(\mathbb{E}[\omega_h] - \omega_k^*) \right]\rangle$$

$$\leq \frac{\mu_F}{4} \| \mathbb{E}[\omega_h] - \omega_k^*\|^2 + \frac{1}{\mu_F} \left\| \mathbb{E}[\nabla_\omega \tilde{L}_h] - F_\rho(\theta_k)(\mathbb{E}[\omega_h] - \omega_k^*) \right\|^2$$

$$\leq \frac{\mu_F}{4} \| \mathbb{E}[\omega_h] - \omega_k^*\|^2 + \frac{3}{\mu_F} \left[ \hat{\delta}_F^2 \left\{ \mathbb{E}\left[ \|\omega_0 - \omega_k^*\|^2 \right] + \alpha R_0 + R_1 \right\} + \frac{\mu_F^{-2}G^2}{(1-\gamma)^4}\hat{\delta}_F^2 + \hat{\delta}_g^2 \right]$$

Substituting the above bounds in (63), we obtain the following recursion.

$$\| \mathbb{E}[\omega_{h+1}] - \omega_k^* \|^2 \leq \left( 1 - \frac{3\alpha\mu_F}{2} + 2G^4\alpha^2 \right) \| \mathbb{E}[\omega_h] - \omega_k^* \|^2$$
$$+ 6\alpha \left( \alpha + \frac{1}{\mu_F} \right) \left[ \hat{\delta}_F^2 \left\{ \mathbb{E}\left[ \|\omega_0 - \omega_k^*\|^2 \right] + \alpha R_0 + R_1 \right\} + \frac{\mu_F^{-2}G^2}{(1-\gamma)^4} \hat{\delta}_F^2 + \hat{\delta}_g^2 \right]$$

If $\alpha < \frac{\mu_F}{4G^4}$, the above bound implies the following.

$$\| \mathbb{E}[\omega_{h+1}] - \omega_k^* \|^2 \leq (1 - \alpha\mu_F) \| \mathbb{E}[\omega_h] - \omega_k^* \|^2$$
$$+ 6\alpha \left( \alpha + \frac{1}{\mu_F} \right) \left[ \hat{\delta}_F^2 \left\{ \mathbb{E}\left[ \|\omega_0 - \omega_k^*\|^2 \right] + \alpha R_0 + R_1 \right\} + \frac{\mu_F^{-2}G^2}{(1-\gamma)^4} \hat{\delta}_F^2 + \hat{\delta}_g^2 \right]$$

Unrolling the above recursion, we obtain

$$\| \mathbb{E}[\omega_H] - \omega_k^* \|^2 \leq (1 - \alpha\mu_F)^H \| \mathbb{E}[\omega_0] - \omega_k^* \|^2$$
$$+ \sum_{h=0}^{H-1} 6 (1 - \alpha\mu_F)^h \alpha \left( \alpha + \frac{1}{\mu_F} \right) \left[ \hat{\delta}_F^2 \left\{ \mathbb{E}\left[ \|\omega_0 - \omega_k^*\|^2 \right] + \alpha R_0 + R_1 \right\} + \frac{\mu_F^{-2}G^2}{(1-\gamma)^4} \hat{\delta}_F^2 + \hat{\delta}_g^2 \right]$$
$$\leq \exp\left( -H\alpha\mu_F \right) \| \mathbb{E}[\omega_0] - \omega_k^* \|^2 + \frac{6}{\mu_F} \left( \alpha + \frac{1}{\mu_F} \right) \left[ \hat{\delta}_F^2 \left\{ \mathbb{E}\left[ \|\omega_0 - \omega_k^*\|^2 \right] + \alpha R_0 + R_1 \right\} + \frac{\mu_F^{-2}G^2}{(1-\gamma)^4} \hat{\delta}_F^2 + \hat{\delta}_g^2 \right]$$

This concludes the result.

## F. Proof of Theorem 3

Combining Lemma 6 with Lemma 3, we get the following bound for appropriate choices of the learning rates as mentioned in the theorem.

$$J_\rho^* - \frac{1}{K} \sum_{k=0}^{K-1} \mathbb{E}[J_\rho(\theta_k)] \leq \sqrt{\epsilon_{\text{bias}}} + G \exp\left( -\frac{H\alpha\mu_F}{2} \right) \| \mathbb{E}[\omega_0] - \omega^* \|^2 + C_2$$
$$+ \frac{B}{4L} \left( \frac{\mu_F^2}{G^2} + G^2 \right) \exp\left( -H\alpha\mu_F \right) \mathbb{E} \|\omega_0 - \omega^*\|^2 + C_3 + C_4 \tag{64}$$

with $C_2 = G\sqrt{C_1}$, $C_3 = \frac{B}{4L} \left( \frac{\mu_F^2}{G^2} + G^2 \right) C_0$ and $C_4 = \frac{G^2}{\mu_F^2 K} \left( \frac{B}{1-\gamma} + 4L \mathbb{E}_{s \sim d_\rho^{\pi^*}} [KL(\pi^*(\cdot|s) \| \pi_{\theta_0}(\cdot|s))] \right)$. Where $C_0$ and $C_1$ are defined in Lemma 6. By replacing $C_4 = \mathcal{O}\left( \frac{1}{K} \right)$ we can express (64) as follows:

$$J_\rho^* - \frac{1}{K} \sum_{k=0}^{K-1} \mathbb{E}[J_\rho(\theta_k)] \leq \sqrt{\epsilon_{\text{bias}}} + \mathcal{O}(e^{-H}) + \mathcal{O}\left( \gamma^{2N} + \frac{\hat{\sigma}_F^2}{(1-\gamma)^4} + \hat{\sigma}_g^2 \right) + \mathcal{O}\left( \frac{1}{K} \right) \tag{65}$$

To obtain the optimality gap at most $\sqrt{\epsilon_{\text{bias}}} + \epsilon$, we choose $H = \mathcal{O}(\log(\epsilon^{-1}))$, $N = \mathcal{O}(\log(\epsilon^{-1}))$, $K = \mathcal{O}(\epsilon^{-1})$, $\hat{\sigma}_g^2 = \mathcal{O}(\epsilon)$ and $\hat{\sigma}_F^2 = \mathcal{O}((1-\gamma)^4\epsilon)$. By Theorem 2, $\tilde{\mathcal{O}}\left( \frac{G^2 d}{(1-\gamma)^2\sqrt{\epsilon}} \right)$ queries of $\mathcal{U}_F$ and $\tilde{\mathcal{O}}\left( \frac{Gd}{(1-\gamma)^2\sqrt{\epsilon}} \right)$ queries of $\mathcal{U}_g$. This results in a sample complexity of $\tilde{\mathcal{O}}\left( HKN \cdot \frac{d}{(1-\gamma)^2\sqrt{\epsilon}} \right) = \tilde{\mathcal{O}}(1/\epsilon^{1.5})$ and an iteration complexity of $\mathcal{O}(K) = \tilde{\mathcal{O}}(1/\epsilon)$.

