# OpenReview forum: "Accelerating Quantum Reinforcement Learning with a Quantum Natural Policy Gradient Based Approach"
_purdue.edu/Purdue_University/PQAI/2025/Symposium — PQAI 2025 Oral_

### Official Review · Reviewer_s47S · 2025-07-19
**Nice theoretical contribution**

**Rating:** 7
**Confidence:** 4

**Review:**

In my opinion, the work makes a valuable contribution by demonstrating a quantum advantage for a specific case.
The work is very carefully elaborated.
The work is purely theoretical without empirical evidence. This is somewhat of a pity, but it is better to work out one step carefully than to do two sloppily.

Further comments:

Some abbreviations are introduced several times in the text, it is better to introduce them only once and then use them consistently. Examples:

Markov Decision Process (MDP)

Natural Policy Gradient (NPG)

At „Quantum Reinforcement Learning: In recent years, Quantum Reinforcement Learning (QRL) has garnered substantial interest from the research community (Dong et al., 2008; Paparo et al., 2014; Dunjko et al., 2017; Jerbi et al., 2021)“, the survey [1] could also be referenced, which gives the reader a good overview. More recently offline QRL has been introduced in the model-free [2] and model-based [3] setting.

In „policy in (4)-(6)“, it looks strange that “in” is blue.

“assumption 1” -> “Assumption 1”

In the bibliography, care must be taken that the automatic lower case is suppressed in some cases by curly brackets, e.g. “marov”, “gaussian”, ‘grover’

In the equation in the appendix at “Load the initial state”, the use of s0 and S0 is somewhat unfortunate because both look so extremely similar.

[1] N. Meyer et al., A survey on quantum reinforcement learning, 2022

[2] Z. Cheng et al., Offline Quantum Reinforcement Learning in a Conservative Manner, 2023

[3] S. Eisenmann et al., Model-based Offline Quantum Reinforcement Learning, 2024

Recommendation:

Accept (Poster)

---

### Official Review · Reviewer_aBuj · 2025-07-23
**A well-theorized and novel approach to model-free quantum reinforcement learning that improves sample efficiency, though limited by idealized assumptions and the absence of empirical results—overall acceptable.**

**Rating:** 7
**Confidence:** 4

**Review:**

This paper tackles model-free quantum reinforcement learning using oracle access and proposes a quantum-compatible version of Natural Policy Gradient. By replacing random sampling with a deterministic gradient estimation and embedding it into quantum states, the method improves sample efficiency over classical approaches. The theory is solid and well-analyzed, and while it relies on idealized assumptions and lacks experimental results, the contribution is novel and promising.

---

### Official Review · Reviewer_kjai · 2025-07-23

**Rating:** 8
**Confidence:** 4

**Review:**

This paper studies the quantum algorithm for reinforcement learning. It proposes a quantum natural gradient descent algorithm and achieves a sample complexity $\tilde{O}(\epsilon^{-1.5})$, improving the classical state-of-the-art result of $\tilde{O}(\epsilon^{-2})$. Technically, it mainly relies on the quantum mean estimation and variance reduction, which were previously developed in (Sidford & Zhang, 2024) for a different problem. Nevertheless, this is the first quantum work to address infinite-horizon MDPs with general parameterized policies.

Overall, this paper advances our understanding of using quantum computers to accelerate classical machine learning algorithms. The results are solid and could be interesting to the QML audience. Therefore, I recommend accepting it as oral.

Question:
* In the abstract, it mentions that the classical $\tilde{O}(\epsilon^{-2})$ is a lower bound. However, in the main text, it appears that this is only an upper bound achieved by the currently best classical algorithm. It should be clarified in the paper. If there is a provable lower bound for this problem, the notation should be $\tilde{\Omega}(\epsilon^{-2})$.

---

### Decision · Program_Chairs · 2025-07-29

Accept (Oral)